# SuperHype: Hypergraph Generation via Graph-Superposition Decomposition

## Abstract

Hypergraphs are graph generalizations with key applications in domains such as healthcare, where strict data privacy requirements apply, or bioinformatics, where testing new compounds is costly. However, research into hypergraph synthesis is limited, and state-of-the-art approaches yield limited generation quality in terms of overall structural patterns and graph-level validity. This is caused by the hypergraph's combinatorial structure, which is composed of a number of possible hyperedges that is factorial in the number of nodes. In fact, current solutions rely on diffusion models denoising graph projections, which are exact but inefficient, or lightweight but approximate. To address such shortcomings, we introduce SuperHype, the first hypergraph diffusion model with tractable and exact modeling. To tackle the complexity of hypergraph representation, we introduce graph superposition, a novel representation that embeds a hypergraph into a multilayer graph. Superposition enables a tractable representation that maintains exactness. To generate new samples from such representations, we introduce a Graph-Superposition Transformer that treats the superposition as an interconnected sequence of layers. We optimize the model architecture to learn low-level patterns within individual graphs in the superposition and high-level patterns between the different graphs of the same superposition. Moreover, we enhance the model's performance with hypergraph-specific auxiliary features and triplet aggregation of indirect node interactions. Our evaluation on five datasets shows that SuperHype generally reproduces local and global connectivity patterns with superior fidelity to state-of-the-art baselines.

## 1 Introduction

Hypergraphs can model intricate, high-order relationships across diverse domains, such as social networks, bioinformatics, and recommender systems, and are used on tasks such as node classification and link prediction (Feng et al., 2025; Bazaga et al., 2024). To train such models on sensitive inputs, synthetic data produced via generative models has emerged as a prominent solution for various data types, such as images (Suh et al., 2024), tables (Kotelnikov et al., 2023), and including hypergraphs Wang et al. (2018); Wen & Yu (2025); Gailhard et al. (2025). However, such state-of-the-art hypergraph generators remain severely limited. These recently proposed hypergraph synthesizers rely on architectures that are unsuitable for the characteristics of hypergraphs and offer limited generative capabilities.

The main challenge in synthesizing hypergraphs lies in the complexity of their representation. While traditional graphs with $\mathcal{V}$ nodes can be efficiently represented in $\mathcal{O}(\mathcal{V}^2)$ via adjacency matrices, the number of possible hyperedges in a hypergraph is $\mathcal{O}(2^\mathcal{V})$. A common representation paradigm is the *bipartite representation*, which embeds a hypergraph with $\mathcal{V}$ nodes and $\mathcal{E}$ edges as a bipartite graph with $\mathcal{V} + \mathcal{E}$ nodes, in which original nodes are connected to the hyperedges containing them. Other representation paradigms such as the *clique expansion* are tractable but introduce ambiguity in their embeddings. Hence, one representation can belong to multiple hypergraphs. Figure 1 showcases these problems.

To enable the generation of synthetic hypergraphs without sacrificing the rich structural information they contain, we propose SuperHype, the first *diffusion model for hypergraph generation*. SuperHype makes hypergraph synthesis tractable via a **graph superposition projection**, a novel algo-

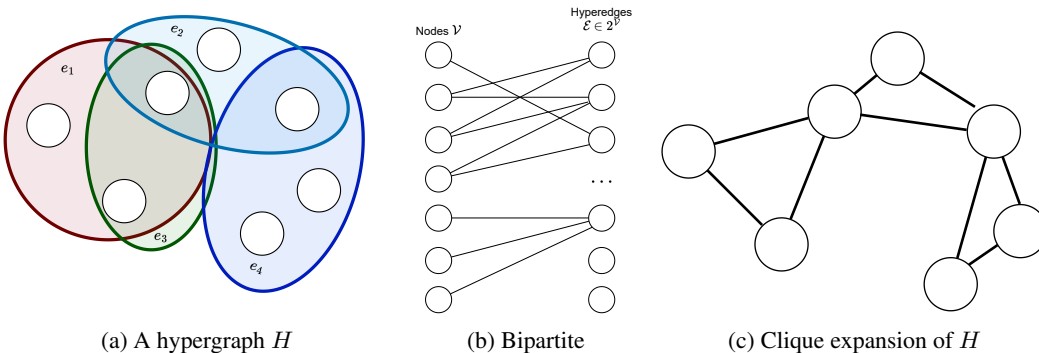

(a) A hypergraph $H$       (b) Bipartite       (c) Clique expansion of $H$

Figure 1: Hypergraph representation is an NP-complete problem. (a) A hypergraph $H$; (b) The bipartite representation requires an extensive enumeration of the $\mathcal{O}(2^{\mathcal{V}})$ possible edges; (c) Using the clique expansion, the presence of the edge $e_3$ is lost due to ambiguity.

rithm that embeds a hypergraph into a multi-layer graph, introducing tractability while retaining representation unambiguity. Unlike classical clique expansion, the mapping from a graph superposition to a hypergraph is injective, which means that the projection into a graph superposition does not induce a loss of information. To handle these representations in the context of diffusion models, we propose a **graph-superposition transformer**, a transformer-based diffusion model that performs message passing both within individual graph layers and across layers in the superposition. Coupled with a discrete denoising-diffusion process, the model produces high-fidelity synthetic hypergraphs that faithfully reproduce real-world patterns. Finally, for sharper control over maximal-clique formation, we augment the model with a **triplet aggregation mechanism** and specific auxiliary features. Together, these additions markedly improve the accuracy of the generated hypergraphs' local motifs and global topology.

We evaluate SuperHype generative performance against 6 state-of-the-art baselines on 5 different datasets and demonstrate that SuperHype consistently achieves superior synthesis quality. Our contributions are as follows:

- SuperHype introduces a compact and exact representation for hypergraphs, along with a greedy algorithm to construct such representations.
- SuperHype introduces the first graph-superposition transformer. A novel neural network architecture for graph diffusion that enables high-fidelity hypergraph synthesis with tractable cost.
- SuperHype is enriched with sharp control over maximal-clique formation, leading to better synthesis quality.
- Our experiments over 5 hypergraph datasets demonstrate that SuperHype consistently achieves superior generation quality against 6 state-of-the-art baseline generators.

Our code is available anonymously at:
`https://anonymous.4open.science/r/SuperHype-918F`

## 2 RELATED WORK

**Graph Generative Models** Graph generation has been at the forefront of scientific research, with applications ranging from molecule design to social-network simulation. Early deep generative approaches —based on variational autoencoders (Simonovsky & Komodakis, 2018), GANs (Wang et al., 2018), or autoregressive schemes (You et al., 2018)— struggled with graph-specific constraints like permutation equivariance, discreteness, and sparsity. More recent diffusion-based formulations better address these challenges Vignac et al. (2023); Jo et al. (2024).

Diffusion models have emerged as powerful generative frameworks across continuous and discrete data modalities. *Denoising Diffusion Probabilistic Model* (DDPM) (Ho et al., 2020) defines a

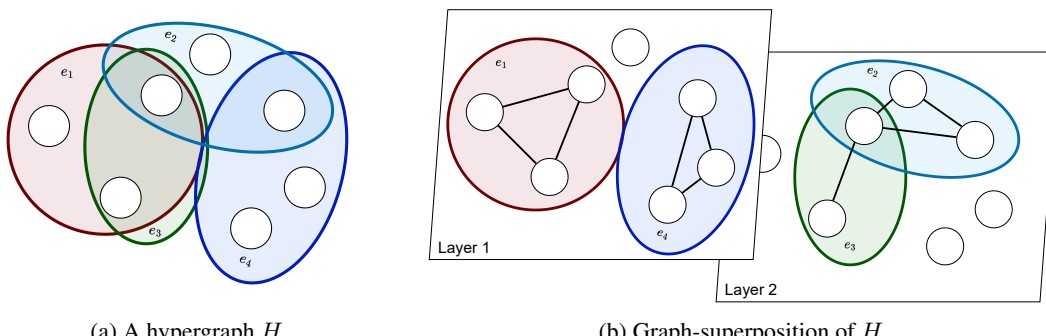

(a) A hypergraph $H$.  (b) Graph-superposition of $H$.

Figure 2: The graph-superposition is a tractable representation that retains unambiguity. (a) A hypergraph $H$ with 7 nodes and 4 hyperedges; (b) A graph-superposition of $H$ on 2 layers: retains exactness while being tractable.

Markov chain that iteratively injects noise in the data, i.e., forward process. A neural network is trained to reverse this process, enabling generation from a simple, tractable distribution such as Gaussian noise. When applying diffusion models to graphs, both discrete and continuous approaches are proposed. On the one hand, *DiGress* (Vignac et al., 2023) introduces a denoising diffusion process that preserves the discrete, sparse nature of adjacency matrices and proposes a graph transformer with auxiliary graph properties. On the other hand, *GruM* (Jo et al., 2024) models global topology as a mixture of diffusion processes, while *Cometh* (Siraudin et al., 2024) allows flexible control of the denoising schedule to trade off synthesis quality and computation. These advances yield markedly improved graph connectivity and realism, though often at higher computational cost.

Nevertheless, while graph diffusion models can provide a solid base for the hypergraph setting, existing models are unfit for generating such richer structures due to their explicit reliance on simple two-node edges and, often, the adjacency matrix format.

**Hypergraph Generative Models** Hypergraphs generalize graphs by allowing *hyperedges* to connect more than two nodes, enabling richer modeling of higher-order relationships. Classical approaches infer hypergraphs from graphs, for example, through Bayesian reconstruction (Young et al., 2020) or machine learning methods trained to invert clique expansion (Wang & Kleinberg, 2024). GAN-based approaches (Pan et al., 2021) and variational autoencoders (Su et al., 2024) offer initial solutions, while HyperPLR (Wen & Yu, 2025) projects hypergraphs into weighted clique graphs before reconstruction via greedy algorithms. Nevertheless, HyperPLR generates a clique expansion of a hypergraph and then tries to convert it into a hypergraph using a greedy algorithm. This transformation induces a loss of information in the generated hypergraph. More recently, HYGENE (Gailhard et al., 2025) directly applies denoising diffusion to hypergraph generation, using iterative extrapolation. However, extrapolation of graphs yields suboptimal generation quality. In summary, the current state-of-the-art falls short in capturing specific hypergraph properties, leading to limited synthesis quality.

## 3    SUPERHYPE

We start by going over preliminaries and notation for our hypergraph generation problem in Section 3.1. Then, we introduce our first main contribution in Section 3.2: *superposition decomposition* for hypergraphs. To enable generation over this newly introduced graph superposition representation, which takes the form of a small collection of related graphs, we extend discrete graph diffusion models into the *Graph-Superposition Transformer* as described in Section 3.3. Finally, to further improve the performance of SuperHype, we augment our model with *hypergraph-specific auxiliary features* and *triplet aggregation*, enhancing its ability to model higher-order graph properties and interactions between nodes with a common neighbor.

## 3.1 Preliminaries

While the set of possible edges in a graph $G = (\mathcal{V}, E)$ is $\mathcal{O}(V^2)$, the set of all possible hyperedges in a hypergraph $H = (\mathcal{V}, \mathcal{E})$, with nodes $\mathcal{V} \in \mathbb{N}$ and hyperedges $\mathcal{E} \subset 2^{\mathcal{V}}$, is $\mathcal{O}(2^{\mathcal{V}})$. Hence, representing hypergraphs is NP-complete. A naive way of representing hypergraphs is as a *bipartite graph* with a set of nodes $\mathcal{V} \cup \mathcal{E}$, in which original nodes are connected with the hyperedges they belong to. This representation is extensive but introduces prohibitive costs as it scales as $\mathcal{O}(2^{\mathcal{V}})$.

**Clique expansion** To mitigate the prohibitive cost of hypergraph representation, a mechanism based on *clique expansion* is used.

**Definition 3.1** (Clique expansion of a hypergraph). Let $H = (\mathcal{V}, \mathcal{E})$ be a hypergraph. Its *clique expansion* is the graph $G = (\mathcal{V}', \mathcal{E}')$ satisfying the following conditions:

    **i. Same set of vertices**: $\mathcal{V}' = \mathcal{V}$.

    **ii. Adjacency via hyperedge**: $(\{i, j\} \in \mathcal{E}' \iff \exists e \in \mathcal{E}, \{i, j\} \subseteq e), \forall i, j \in \mathcal{V}^2$.

The clique expansion of a hypergraph defined in Definition 3.1 transforms the higher-order relationships of hyperedges into pairwise connections. Consequently, it induces a loss of information in the generated graph. Indeed, a clique-expansion representation may correspond to multiple hypergraphs, because every subset of a clique remains a clique, whereas a subset of a hyperedge does not necessarily form a hyperedge.

## 3.2 Superposition decomposition

Intractability of hypergraph representation prevents its application to diffusion models, whose computational and memory costs scale in the size of the data. Hence, current methodologies for hypergraph synthesis limit these costs by either relying on graph extrapolation or ambiguous hypergraph representation. However, this unfortunately degrades the generation quality of their models.

To overcome this limitation, we develop a **hypergraph superposition decomposition**, which enables tractable and unambiguous hypergraph representation. By decomposing the hypergraph into multiple layered subgraphs, SuperHype manages to reduce the computational cost of hypergraph representation from $\mathcal{O}(\mathcal{VE})$ to $\mathcal{O}(\mathcal{V}^2 L)$, where $L$ is the number of layers used to store the graph. Here, the challenge is twofold: minimizing *memory cost* and guaranteeing *unambiguity*. First, since the number of total edges is $\mathcal{O}(2^{\mathcal{V}})$, they must be spread over the layers as much as possible to contain the memory costs. Secondly, the hyperedges must be distributed to avoid ambiguity, allowing for exact reconstruction. To overcome these challenges, we propose a greedy algorithm that maps a hypergraph $H = (\mathcal{V}, \mathcal{E})$ to a *graph superposition representation* $(\mathcal{V}, (\mathcal{E}_l)_{l=1}^L)$ in $O(3^{\mathcal{V}/3} L \mathcal{V})$. We formalize the graph superposition representation of a hypergraph in Definition 3.3.

---

**Algorithm 1:** Graph superposition projection

**Input:** Hypergraph $H = (\mathcal{V}, \mathcal{E})$,
enumeration $(e_i)_{i=1}^{|\mathcal{E}|}$ of elements in $\mathcal{E}$,
number of layers $L$
**Output:** Graph-superposition $(\mathcal{V}, (\mathcal{E}_l)_{l=1}^L)$
$(\mathcal{E}_l)_{l=1}^L \leftarrow (\emptyset)_{l=1}^L$
**for** $i \leftarrow 1$ **to** $|\mathcal{E}|$ **do**
    $\pi \leftarrow \text{RandPerm}(1, \ldots, L)$
    **for** $l' \leftarrow 1$ **to** $L$ **do**
        $l \leftarrow \pi[l']$
        $\mathcal{E}' \leftarrow \mathcal{E}_l \cup \mathcal{P}_2(e_i)$
        $\mathcal{C} \leftarrow \text{MaximalCliques}(\mathcal{V}, \mathcal{E}_l)$
        $\mathcal{C}' \leftarrow \text{MaximalCliques}(\mathcal{V}, \mathcal{E}')$
        **if** $e_i \notin \mathcal{C} \wedge \mathcal{C}' = \mathcal{C} \cup \{e_i\}$ **then**
            $\mathcal{E}_l \leftarrow \mathcal{E}'$
            **break**
        **end**
        **else**
            **fail**
        **end**
    **end**
**end**
**return** $(\mathcal{V}, (\mathcal{E}_l)_{l=1}^L)$

---

**Definition 3.2** (Graph superposition). Let $l \in \mathbb{N}^*$. A *graph superposition* $(\mathcal{V}, (\mathcal{E}_i)_{i=1}^l)$ with $L$ layers is a $L$-tuple of graphs, each with nodes $\mathcal{V}$. For each $l \in [\![1, L]\!]$, $(\mathcal{V}, \mathcal{E}_l)$ is a graph.

**Definition 3.3** (Graph-superposition representation of a hypergraph). Let $H = (\mathcal{V}, \mathcal{E})$ be a hypergraph. A *graph-superposition representation* of $H$ is a graph superposition $(\mathcal{V}, (\mathcal{E}_l)_{l=1}^L)$ sharing the

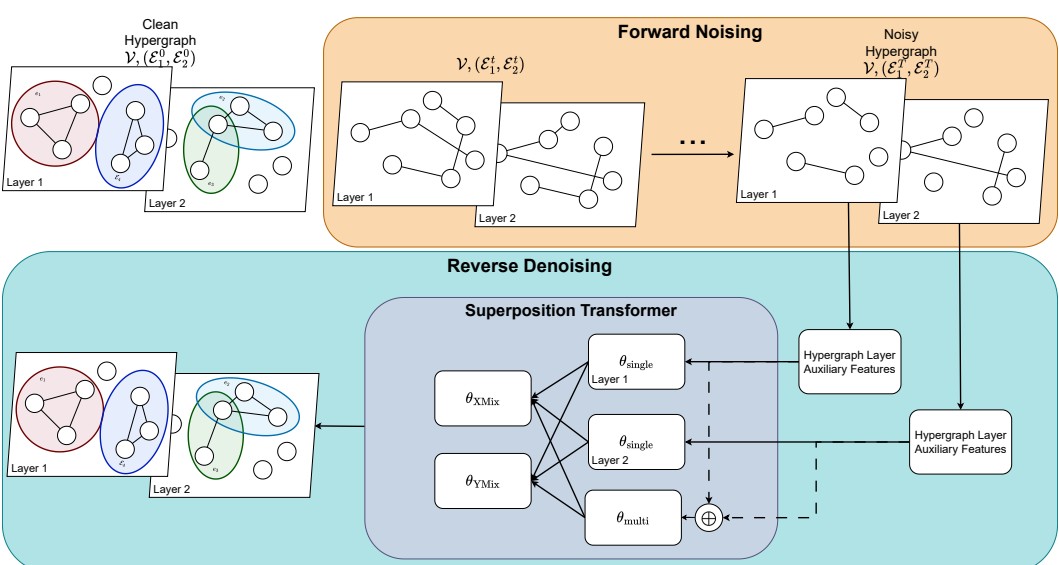

Figure 3: SuperHype Diffusion Process and Model Architecture Overview

same node set $\mathcal{V}$ as $H$ so that the hyperedge set $\mathcal{E}$ of $H$ is the disjoint union of the maximal clique sets of the graph layers: $\mathcal{E} = \bigsqcup_{l=1}^{L} \mathcal{M}(\mathcal{E}_l)$, where $\mathcal{M}(\mathcal{E}_l)$ is the set of all maximal cliques in $\mathcal{E}_l$.

**Graph superposition projection** We build a greedy algorithm to generate a graph superposition from a hypergraph $H = (\mathcal{V}, \mathcal{E})$ in $O(3^{n/3}LN)$. The procedure is outlined in Algorithm 1. The generation of the graph-superposition projection relies on an algorithm that sequentially adds hyperedges. It starts with an empty graph-superposition representation with a predefined number of layers. This graph superposition is filled progressively by adding hyperedges one by one in a random order. For every new hyperedge to add, a random layer is chosen from those that do not interfere with the new maximal clique to add, following Definition 3.4. If the hyperedge cannot be added to any layer, the algorithm fails. After a predefined number of attempts, the hypergraph is considered non-projectable on the given number of layers.

**Definition 3.4** (Condition to add a hyperedge to a layer). Let $(\mathcal{V}, (\mathcal{E}_l)_{l=1}^{L})$ be a graph-superposition representation and $e$ a hyperedge. For a given layer number $l \in [\![1, L]\!]$, let $\mathcal{M}$ be the set of the maximal cliques in the graph $(\mathcal{V}, \mathcal{E}_l)$ and $\mathcal{M}'$ the set of the maximal cliques in the graph $(\mathcal{V}, \mathcal{E}_l \cup \mathcal{P}_2(e))$. $\mathcal{P}_2(e) \subset \mathcal{V}^2$ is the set of all possible node pairs part of hyperedge $e$. The hyperedge $e$ is admissible for addition to the $l$-th layer of the graph superposition if the following conditions are satisfied:

$$e \notin \mathcal{E}_i \quad \wedge \quad \mathcal{M}' = \mathcal{M} \cup \{e\} \tag{1}$$

**Hypergraph reconstruction** Given a graph-superposition, the corresponding hypergraph can be reconstructed with no loss of information using the maximal cliques of each graph layer. The algorithm starts with the empty hypergraph $H = (\mathcal{V}, \emptyset)$. For each layer of the superposition, the set of maximal cliques is $\mathcal{M} = \{\mathcal{E}_1, \ldots, \mathcal{E}_m\}$. Each of the cliques $\mathcal{E}_i$ is then used to form a new hyperedge and added to $H$. At the end of this procedure, $H$ is the same hypergraph used to form the graph-superposition.

### 3.3 DIFFUSION VIA SUPERPOSITION TRANSFORMER

Since the proposed superposition transformation encompasses multiple standard graphs, we require an extended graph diffusion model formulation to generate new samples. Moreover, alongside dependencies between different nodes and edges within a single graph structure, our model must consider dependencies between different representations of the same node at different superposition layers, and between the graph-level properties of the different layers. Thus, we present the generalized diffusion technique, followed by details of our novel transformer architecture below. Figure 3 shows the diffusion pipeline and the integration of our graph-superposition transformer-based model.

**Forward diffusion:** We use discrete graph diffusion (Vignac et al., 2023) as the starting point for our diffusion process. In the case of an unattributed graph with binary edges as input, the classic forward noising process is modeled as a Markov Chain. Within it, the adjacency matrix encoding the edges of a graph is progressively corrupted towards a predetermined prior distribution over a specified number of time steps, based on a noise schedule. Notably, a key feature of discrete diffusion is that the edges in the adjacency matrices at different steps remain discrete, yielding valid graph structures. In our case, given the superposition $(\mathcal{V}, (\mathcal{E}_l)_{l=1}^d)$, let $(\mathcal{E}_l^t)_{ij} \in \{0,1\}$ denote the adjacency entry of nodes $i$ and $j$ in superposition layer $1 \leq l \leq L$ at time step $0 \leq t \leq T$. Time step $t = 0$ corresponds to the original graph, while the case $t = T$ corresponds to the fully noised graph. For each layer $l$, we have the following transition probability relationship from the single graph case:

$$\begin{bmatrix} P\left((\mathcal{E}_l^{t+1})_{ij} = 1\right) \\ P\left((\mathcal{E}_l^{t+1})_{ij} = 0\right) \end{bmatrix} = \begin{bmatrix} \alpha_t & m(1-\alpha_t) \\ (1-m)(1-\alpha_t) & \alpha_t \end{bmatrix} \begin{bmatrix} P\left((\mathcal{E}_l^t)_{ij} = 1\right) \\ P\left((\mathcal{E}_l^t)_{ij} = 0\right) \end{bmatrix}$$

where $(\mathcal{E}_l^{t+1})_{ij}$ is the adjacency matrix entry for the edge $i - j$ in $\mathcal{E}_l^{t+1}$, $\alpha^t$ is set according to the noise schedule, while $m$, as the marginal probability of an edge existing, gives the noise distribution. We keep the noise process independent per edge, also across layers, to maintain a tractable process, which is also fast to apply for efficient training (and sampling).

**Reverse diffusion:** The reverse diffusion process involves iteratively denoising a random sample from the noise distribution without having access to the reference clean graph, effectively generating new data points from the modeled distribution. Specifically, we train a neural network $\theta$ to approximate the clean graph from any given time step $t$. We gradually denoise the sampled noise graph during generation based on $\theta$'s prediction over $T$ steps to converge toward a realistic graph sample. To train the model, we treat each adjacency matrix entry in each superposition layer as a binary classification problem, and thus optimize a cross entropy objective between the probability distribution predicted by the model for each edge and the ground truth edge values. Specifically, for a reference clean graph superposition $(\mathcal{E}_l^0)_{l=1}^L$, and the prediction $(\hat{\mathcal{E}}_l^0)_{l=1}^L$, we have:

$$\mathcal{L} = \sum_{l=1}^L \frac{2}{\mathcal{V}^2} \sum_{i,j \in \mathcal{V}^2} \text{CrossEntropy}((\hat{\mathcal{E}}_k^0)_{ij}, \ (\mathcal{E}_k^0)_{ij})$$

**Graph-Superposition Transformer** To take advantage of the richer representation enabled by the superposition, we must extend the standard graph transformer architecture at the core of many modern graph diffusion models to account for interaction between superposition layers. As the superposition layers share the same set of nodes, the node representations at different layers should inform each other. Similarly, it is beneficial to share graph-level information between the layers. For edges, we do not want to merge edge representations between layers at an input level, as that would destroy one of the main characteristics of the superposition. However, we share the same set of neural network weights for edges across layers, so common patterns across layers are modeled more efficiently and do not have to be relearned separately. While we strictly generate structural information for our task, previous discrete diffusion work has shown that adding auxiliary node, graph, and edge-level information to the model input yields significantly increased generation quality Xu et al. (2024); Vignac et al. (2023). Such information is derived directly from the input graph, in the form of node degrees, cycle counts, or the total number of connected components for the whole graph. Our modified architecture relies heavily on such auxiliary structural features, and in Section 3.4 we discuss specific hypergraph-specific auxiliary features we use to maximize performance.

Each transformer block in our model incorporates five different types of embeddings created from auxiliary features (and, in the case of edges, additionally, describing some connectivity pattern). Figure 3 includes an overview of such a transformer block. We have three layer-specific embeddings: node $\mathbf{X}$, edge $\mathbf{E}$, and graph $\mathbf{Y}$ features. They are only applied to the different components within the same layer, and modeled by a set of parameters $\theta_{\text{single}}$ shared across layers. The two multi-layer embeddings are for: nodes $\mathbf{X}^+ = \sum_{l=1}^L \mathbf{X}_l$ and the whole graph $\mathbf{Y}^+ = \sum_{l=1}^L \mathbf{Y}_l$. We initialize each via the sum of the corresponding embeddings within each layer. We model interaction between $\mathbf{X}^+$ and $\mathbf{Y}^+$ with a set of parameters $\theta_{\text{multi}}$. After the processing via the two parallel parameter sets $\theta_{\text{single}}$ and $\theta_{\text{multi}}$, we also mix the updated global node embeddings $\mathbf{X}^+$ with the ones from different layers $(\mathbf{X}_1, \ldots, \mathbf{X}_L)$ via another set of parameters $\theta_{\text{XMix}}$. We proceed analogously for graph-level embeddings via a $\theta_{\text{YMix}}$ parameter set.

### 3.4 HYPERGRAPH-SPECIFIC AUXILIARY FEATURES AND TRIPLET AGGREGATION

Below, we introduce two optimizations applied to our graph-superposition transformer model.

**Auxiliary structural features**: Since the graph-superposition representation focuses on cliques to describe hypergraphs, it is natural to provide the graph-superposition transformer with information about the cliques in the input. We focus on 3-cliques and 4-cliques, as they can be computed efficiently and remain relevant even for hyperedges containing more than four nodes, since multiple 3- and 4-cliques are contained within larger (maximal) cliques. ~~Moreover, the choice also covers most of the hyperedges (over more than two nodes) observed across the datasets in our evaluation.~~ Within each superposition layer, we attach information about the cliques at the node and edge level, denoting the number of cliques the node/edge is part of, and for the whole graph, denoting the total number of cliques. We include more details in Appendix A.1.

**Triplet aggregation**: As covered above, in a clique, every subset of nodes also forms a clique. Thus, smaller cliques, like 3-cliques, are a fundamental building block for larger cliques. On top of the enriching model input with auxiliary features, we thus tackle the problem directly at the architecture level. Following Hussain et al. (2024), we add triplet aggregation to our graph transformer module $\theta_{\mathrm{single}}$, which performs message-passing between layer-specific embeddings. Triplet aggregation brings new interaction signals within the pairs of nodes within 3-cliques relative to the third node. Furthermore, it allows for better modeling of dependencies between non-connected nodes with a common neighbor. Its runtime complexity of $O(\mathcal{V}^{2.37})$ also allows integrating it into the model without creating a bottleneck. A triplet aggregation embedding $a_{ik}$ is computed for every pair of nodes $\{i, j\}$ on a layer by combining an inward update $a_{ik}^{\mathrm{in}}$ and an outward update $a_{ik}^{\mathrm{out}}$ using an MLP. These embeddings are calculated with the scalars $g_{ij}$ and $b_{ij}$, and vectors $\mathbf{v}_{jk}^{\mathrm{in}}$ predicted with MLPs as below:

$$\mathbf{o}_{ij}^{\mathrm{in}} = \sum_{k=1}^{N} a_{ik}^{\mathrm{in}} \mathbf{v}_{jk}^{\mathrm{in}} \qquad\qquad \mathbf{o}_{ij}^{\mathrm{out}} = \sum_{k=1}^{N} a_{ki}^{\mathrm{out}} \mathbf{v}_{kj}^{\mathrm{out}}$$

$$a_{ik}^{\mathrm{in}} = \mathrm{softmax}_k(b_{ik}^{\mathrm{in}}) \times \sigma(g_{ik}^{\mathrm{in}}) \qquad\qquad a_{ki}^{\mathrm{out}} = \mathrm{softmax}_k(b_{ki}^{\mathrm{out}}) \times \sigma(g_{ki}^{\mathrm{out}})$$

## 4 EXPERIMENTS AND RESULTS

In this section, we compare our model with several state-of-the-art hypergraph generation models over different datasets and metrics. The performance of the model is compared against Wen & Yu (2025), Gailhard et al. (2025), and its baselines.

**Datasets**: Our model has been evaluated on the datasets from Gailhard et al. (2025). Four of them are made out of probabilistic models: Erdős-Rényi, SBM - hypergraphs with a community structure, Hypertree - hypergraphs with a tree structure, and Ego - hypergraphs centered on a single node. The last one comes from the conversion of the class bookshelf of ModelNet40 Wu et al. (2015) to hypergraphs.

**Baselines** We compare our model with six baselines. HYGENE is a diffusion-based model taken from Gailhard et al. (2025) for the generation of small synthetic hypergraphs. We also include its baselines VAE, GAN and Diffusion, in which the incidence matrix of the hypergraphs is directly generated using standard AI models. HyperPA is another baseline taken from Gailhard et al. (2025), which involves creating hypergraphs given a predetermined distribution of node degrees and hyperedge sizes. Finally, HyperPLR is the model from Wen & Yu (2025). It consists of generating hypergraphs using the clique expansion as an intermediate representation.

**Evaluation metrics**: In this section, we focus on five main metrics to evaluate our model. Node number, edge size, and node degree metrics are the Wasserstein distance between the distribution of the predicted batch and the distribution of the reference dataset (Appendix D). See Appendix E for more details on the choice of the reference data split for evaluation. The spectral metric is the quadratic maximum mean discrepancy between the two laplacian eigenvalue distributions. For SBM, Hypertree, and Ego, there is also a validity metric that provides the proportion of correct SBM, Hypertree, and Ego hypergraphs among the generated ones.

| Model | SBM Hypergraphs ($n_{avg} = 31.73$, $std = 0.55$) | | | | | Ego Hypergraphs ($n_{avg} = 109.71$, $std = 10.23$) | | | | | Tree Hypergraphs ($n_{avg} = 32$, $std = 0$) | | | | |
|---|---|---|---|---|---|---|---|---|---|---|---|---|---|---|---|
| | V.U.N.↑ | Node Num ↓ | Node Deg ↓ | Edge Size ↓ | Spectral ↓ | V.U.N.↑ | Node Num ↓ | Node Deg ↓ | Edge Size ↓ | Spectral ↓ | V.U.N.↑ | Node Num ↓ | Node Deg ↓ | Edge Size ↓ | Spectral ↓ |
| HyperPA | 2.5% | 7.5e-2 | 4.1 | 4.1e-1 | 2.7e-1 | 0% | 3.6e1 | 2.6 | 4.2e-1 | 2.4e-1 | 0% | 2.4 | 3.2e-1 | 2.8e-1 | 1.6e-1 |
| VAE | 0% | 3.8e-1 | 1.3 | 1.1 | 2.4e-2 | 0% | 4.8e1 | 8.0e-1 | 1.5 | 1.3e-1 | 0% | 9.7 | 7.2e-2 | 4.8e-1 | 1.2e-1 |
| GAN | 0% | 1.2 | 2.1 | 1.2 | 5.9e-2 | 0% | 6.0e1 | 9.1e-1 | 1.7 | 2.3e-1 | 0% | 6.0 | 1.5e-1 | 4.7e-1 | 8.9e-2 |
| Diffusion | 0% | 1.5e-1 | 1.7 | 1.4 | 3.1e-2 | 0% | 4.5 | 4.0 | 3.0 | 1.9e-1 | 0% | 2.2 | 1.7 | 1.9 | 1.3e-1 |
| HyperPLR | 0 (0)% | 1.0e1 (1) | 1.4 (3e-2) | **0 (0)** | 4.0e-2 (2.1e-2) | 0% | 1.3e1 | 4.7e-1 | 1.8e-1 | 2.2e-2 | 0 (0)% | 0 (0)* | 0 (0)* | 0 (0)* | 0 (0)* |
| HYGENE | 63% | **7.2e-2** | 1.7 | 3.5e-3 | 2.6e-2 | **95.6%** | 9.3e-1 | 1.5e-1 | 4.5e-1 | 5.6e-3 | 79.2% | **0** | 3.2e-2 | 1.0e-1 | 7.9e-3 |
| Ours | **88.3 (0.7)%** | 1.0e-1 (1e-2) | **6.8e-1 (1.6e-1)** | 2.6e-3 (4e-4) | **3.0e-3 (2e-4)** | 50.6% | **3.1e-1** | **7.9e-2** | **8.0e-3** | **1.3e-3** | **90 (3)%** | 4.0e-3 (1.7e-3) | **3.7e-3 (7e-4)** | 3.2e-3 (4e-4) | **3.1e-4 (6e-5)** |

Table 1: Comparison between our method and other baselines for SBM, Ego, and Tree hypergraphs. The values in parentheses represent the standard deviations over three experiments. Baseline results are reported from Gailhard et al. (2025) if available. *For HyperPLR on the tree dataset, generated hypergraphs are identical to the training ones.

| Model | Erdos-Renyi Hypergraphs ($n_{avg} = 32$, $std = 0.07$) | | | | ModelNet40 Bookshelf ($n_{avg} = 119.38$, $std = 68.20$) | | | |
|---|---|---|---|---|---|---|---|---|
| | Node Num ↓ | Node Deg ↓ | Edge Size ↓ | Spectral ↓ | Node Num ↓ | Node Deg ↓ | Edge Size ↓ | Spectral ↓ |
| HyperPA | **0.000** | 5.5 | 1.8e-1 | 1.8e-1 | 8.0 | 7.6 | 4.4e-2 | 4.8e-2 |
| VAE | 1.0e-1 | 2.1 | 5.4e-1 | 3.5e-2 | 4.7e1 | 6.2 | 1.5 | 1.9e-1 |
| GAN | 6.8e-1 | 2.6 | 6.6e-1 | 4.8e-2 | **0.0** | 4.0e2 | 4.6e1 | 4.8e-1 |
| Diffusion | 5.0e-2 | 2.2 | 7.8e-1 | 1.4e-2 | **0.0** | 2.0e1 | 2.3 | 7.9e-2 |
| HyperPLR | 1.7 (4e-2) | 5.6 (2e-1) | 1.3 (6e-3) | 1.2e-1 (2e-3) | 1.2 | **3.0** | **0** | **1.9e-3** |
| HYGENE | 2.6e-2 | 2.0e-1 | 1.4e-1 | 4.7e-3 | 1.7 | 8.2 | 3.1e-2 | 6.1e-2 |
| Ours | 7.2e-3 (1.3e-3) | **1.5e-1 (3e-2)** | **6.0e-3 (3.6e-3)** | **9.2e-4 (1.3e-4)** | 2.4 | 3.2 | 7.6e-2 | 2.9e-2 |

Table 2: Comparison between our proposed method and other baselines for the ER and ModelNet40 hypergraphs. The values in parentheses represent the standard deviations over three experiments. Baseline results are reported from Gailhard et al. (2025) if available.

**Experimental setup**: We ran our experiments on Nvidia H100 NVL and L40S GPUs with a computation time for both the training and the evaluation between one and five days. We tuned the size of the embeddings of our model so that these tasks take approximately the same time as the ones for Gailhard et al. (2025). For HyperPLR, it took only a few minutes. For HyperPA, VAE, GAN and Diffusion, we took the results from Gailhard et al. (2025) since their low accuracy does not require an evaluation protocol as precise as the one we used to compare the other models. These metrics are calculated over a batch of 40 generated hypergraphs and a reference dataset with the same number of hypergraphs. Our experiments are conducted with 1000 generated hypergraphs, and the metrics are calculated using the training dataset as a reference. For the experiments on HYGENE, we fixed the ema rate to 0.9999 because there does not seem to be any clear differences between the possible values. This way, we avoid selecting the best values of a random process with variable outcomes. This explains why our results could be different from those in Gailhard et al. (2025).

**Graph-superposition projection**: We projected the hypergraphs of these five datasets to create the corresponding graph-superposition representations. It required 6 layers for the SBM, 5 for the Erdős-Rényi, 1 for the hypertrees, 3 for the Ego dataset, and 6 for the ModelNet40 Bookshelf.

**Results**: We summarize our results in Tables 1 and 2. Overall, our model outperforms the baselines in most cases in its capability to reproduce with accuracy both the local and global features of the training hypergraphs in the generated batch. In all cases we got 100% of generated hypergraphs that are not isomorphic to any other generated one, or to any hypergraph from the training dataset.

The metrics on the edge sizes and node degrees provide information on the capability of the models to reproduce with precision the local structure of the hypergraphs from the training dataset. On all of the datasets with a low number of nodes, our model is better or similar to Gailhard et al. (2025) on this aspect.

**Table 1** On the the SBM, Ego and Tree datasets, SuperHype consistently outperforms most of the baselines. Notably, SuperHype is always the best method under node degree and spectral distribution. Furthermore, SuperHype generates the most valid graphs on the SBM (88.5%) and Tree (90.8%) datasets. While on the Ego dataset SuperHype falls short on the validity, it outperforms

HYGENE on all other quality metrics. Overall, HyperPLR delivers inconsistent generative quality due to the ambiguity introduced by its clique expansion representation. HYGENE deliver decent generative performance thanks to its extensive bipartite representation. However, it falls behind SuperHype 12 times out of 15 scenarios.

**Table 2** On the Erdos-Renyi dataset,SuperHype continues to deliver state-of-the-art generative quality, being the best performing method on 4 metrics out of 5, and second-best in the remaining one. On the larger ModelNet40 dataset, all methodologies struggle to yield consistent quality. Some baselines, like GAN and Diffusion overfit on some metrics while significantly degrading on others. SuperHype, HyperPLR and HYGENE are the only methods that deliver consistent performance. HyperPLR delivers good generative quality on this dataset with SuperHype being a close second. Overall, SuperHype consistently delivers state-of-the-art generative performance on all datasets. At the same time, state-of-the-art methods are highly inconsistent.

The spectral and validity metrics are representative of the capability of the models to learn and reproduce with precision the global structure of the hypergraphs. In Table 1, it seems that our model outperforms Gailhard et al. (2025) in its capability to reproduce valid SBM hypergraphs and hypertrees. In Table 2, the results are still satisfying on the larger hypergraphs of ModelNet40 Bookshelf.

For the ego dataset, the proportion of valid ego hypergraphs is worse with our model than with Gailhard et al. (2025). This comes from the representation of the hyperedges as maximal cliques. If an edge is missing in a set of nodes that is supposed to be a maximal clique representing a hyperedge, the hyperedge is split in two smaller hyperedges and, in some cases, one of them doesn't contain the ego node, which makes an invalid ego hypergraph.

For the ModelNet40 Bookshelf dataset, none of the tested models seems to generate satisfying hypergraphs. Even though the accuracy on node number and edge size is quite correct, the error on the node degrees for HyperPLR, HYGENE and SuperHype are too large for the generated hypergraphs to be used as hypergraphs derived from the original dataset. Indeed, the average value of the node degree for the training dataset is 7.13, whereas the error on node degrees is 3.17 for SuperHype and 8.21 for Gailhard et al. (2025).

**Limitations**: Even though our model shows good results for the generation of hypergraphs with less than 200-250 nodes, it suffers from scalability issues due to a calculation cost that increases sharply with the number of nodes (see Appendix B for details). For this reason, we had to adapt the size of the embeddings for the larger hypergraphs so that the computation cost remains close to that of Gailhard et al. (2025). For even larger hypergraphs, it will be necessary to adapt the pipeline using sparse graph diffusion models such as Qin et al. (2023).

## 5 CONCLUSION

We introduce SuperHype to address the generation quality issues of existing hypergraph generators that rely on bipartite or weighted clique representations. SuperHype is a novel diffusion model for hypergraphs, which represents hypergraphs exactly, and efficiently via a graph-superposition decomposition. SuperHype maps each hypergraph to a small set of graphs, each with the number of nodes as the original hypergraph. We design a graph-superposition transformer, which learn the global data structure across graph projections in forward and reverse diffusion process. Finally, we further enhance the resulting model's performance with auxiliary features and a customized triplet-aggregation attention mechanism. Our experimental evaluation on five hypergraph datasets shows SuperHype outperforming state-of-the-art baselines across graph and spectral metrics.

**Reproducibility Statement** To ensure the reproducibility of our research, we include the code of our proposed model in an anonymized repository.

**Ethic Statement** Hypergraph synthesis has a broad impact in fields in which modeling complex higher-order interactions is relevant, including social networks, epidemics, and chemical reactions. As a generative model, our work can enhance confidentiality of such data instances, which can be used for tasks such as training machine learning models, and evaluating novel algorithms.

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

## A    DETAILS ON THE MODEL

Below, we elaborate on the auxiliary features we pass on to our model and provide additional details of the model architecture.

### A.1    AUXILIARY FEATURES

Before every forward pass in the network, the model calculates additional information to provide the network with a deeper understanding of the complex interactions between the nodes. The vectors that are calculated are given as input data for the different types of embedding, and then transformed into a real embedding via a multi-layer perceptron. Here is the information that is provided for every type of embedding:

- **e** (edges): The number of 3-cliques and 4-cliques containing the edge
- **Y** (graph-superposition): The number of 3-cliques and 4-cliques in the entire graph superposition
- **X** (multi-layer node): The number of 3-cliques and 4-cliques containing one of the different versions of the node across the layers
- **x** (single-layer node): The concatenation of the vector X containing multi-layer information and of the number of 3-cliques and 4-cliques containing the node in the given layer
- **y** (one layer): The concatenation of the vector Y containing multi-layer information and of the number of 3-cliques and 4-cliques in the layer

Note that this auxiliary information is about every type of 3-cliques and 4-cliques, not necessarily the maximal ones.

This process relies on the calculation, for each layer $l$, of the matrices $(B_{i,j})_{1 \leq i,j \leq n}$ and $(C_{i,j})_{1 \leq i,j \leq n}$ containing respectively the number of 3-cliques and 4-cliques for every edge. The basic consists of calculating, for every triplet and quadruplet, the product of the potential edge values in the adjacency matrix:

$$\forall (i,j,l) \in [\![1,n]\!]^2 \times [\![1,l]\!], \begin{cases} B_{i,j} = \sum_{k=1}^{n} A_{i,k} A_{k,j} A_{j,i} \\ C_{i,j} = \sum_{k=1}^{n} \sum_{p=1}^{n} A_{i,k} A_{k,j} A_{j,i} A_{i,p} A_{k,p} A_{j,p} \end{cases} \tag{2}$$

Another approach for these calculations is to take into account the sparsity of the adjacency matrix to avoid making all the products of the edges. The following algorithm reduces the number of operations by skipping the pairs of nodes that have been detected as not being connected:

In Algorithm 2, the loops are interrupted when two nodes in the potential clique are detected as not being connected, which means that the node set has no chance of being a clique.

At the first steps of the denoising process, the distribution of the edges doesn't follow a precise structure. This can sometimes give an enormous number of 3-cliques and 4-cliques that don't really provide the network with usable information on the structure of the hypergraph. Therefore, we define a maximum from which the value is replaced with -1. In our settings, this value is 100 for the edges, 1000 for the nodes, and 10,000 for the entire layers.

This formulation does not use hard combinatorial constraints, even though it could theoretically further boost generation quality, especially for large hyperedges. From experimentally observed hyperedge size and node degree metrics, we find that our hypergraph-specific auxiliary features are nevertheless effective in guiding the model to create cliques. Furthermore, we can theoretically demonstrate that a small number of erroneous edges in the graph-superposition decomposition yields a limited difference in the final number of hyperedges. Take a graph with a single $N$-node maximal clique. If we remove a single edge from this clique, we get two maximal cliques of size $N-1$. Now, let $G$ be an arbitrary graph with an edge $e$ contained in $C$ maximal cliques. Also, let $G'$ be the graph

---

**Algorithm 2:** Optimized version of the algorithm to compute the number of 3-cliques and 4-cliques for every edge in a graph

---

**Input:** Adjacency matrix $(A_{i,j})_{1 \leq i,j \leq n}$ of the $l$-th layer of the graph superposition
**Output:** The matrices $(B_{i,j})_{1 \leq i,j \leq n}$ and $(C_{i,j})_{1 \leq i,j \leq n}$ containing respectively the number of 3-cliques and 4-cliques for every edge.

$(B_{i,j})_{i,j} \leftarrow (0)_{i,j}$
$(C_{i,j})_{i,j} \leftarrow (0)_{i,j}$
**for** $i \leftarrow 1$ **to** $n$ **do**
    **for** $j \leftarrow i + 1$ **to** $n$ **do**
        **if** $A_{i,j} = 0$ **then**
            **break**
        **end**
        **else**
            **for** $k \leftarrow j + 1$ **to** $n$ **do**
                **if** $A_{i,k} = 0 \vee A_{j,k} = 0$ **then**
                    **break**
                **end**
                **else**
                    $B_{i,j} \leftarrow B_{i,j} + 1$
                    **for** $p \leftarrow k + 1$ **to** $n$ **do**
                        **if** $A_{i,p} = 1 \wedge A_{j,p} = 1 \wedge A_{k,p} = 1$ **then**
                            $C_{i,j} \leftarrow C_{i,j} + 1$
                        **end**
                    **end**
                **end**
            **end**
        **end**
    **end**
**end**
**return** $(B_{i,j})_{1 \leq i,j \leq n}$ and $(C_{i,j})_{1 \leq i,j \leq n}$

---

$G$ without edge $e$. Based on the previous reasoning, we can upper bound the difference between the number of maximal cliques $M(G)$ in $G$ and the corresponding $M(G')$ in $G'$ as:

$$|M(G) - M(G')| \leq C$$

Similar reasoning can apply to an edge addition. Thus, even if the denoising process makes a few isolated errors on the edges, the result still corresponds to a hypergraph with a reasonable number of hyperedges.

## A.2 DETAILS ON THE ARCHITECTURE

This section aims to present in detail the network that is used to denoise the graph-superposition representation. In Figure 4, the connections between the different modules of the model are shown.

Figure 5 describes the cross-attention module between local and global embeddings. It corresponds to the XYTransformer ($\theta_{\text{multi}}$ in the main text), the XxTransformer ($\theta_{\text{XMix}}$ in the main text), and the YyTransformer ($\theta_{\text{YMix}}$ in the main text). In all of these cases, the variable L corresponds to the embedding of the local feature, which is a refinement of the global feature G. In the XYTransformer, the local feature is X and the global one is Y. In the XxTransformer, the local feature is x and the global one is X. In the YyTransformer, the local feature is y and the global one is Y.

Figure 6 shows the layout of the graph transformer module, which realizes message-passing between layer-specific embeddings. First, it computes pairwise interactions to update node, edge, and layer embeddings. After this, triplet aggregation coefficients are used to update the edge embeddings.

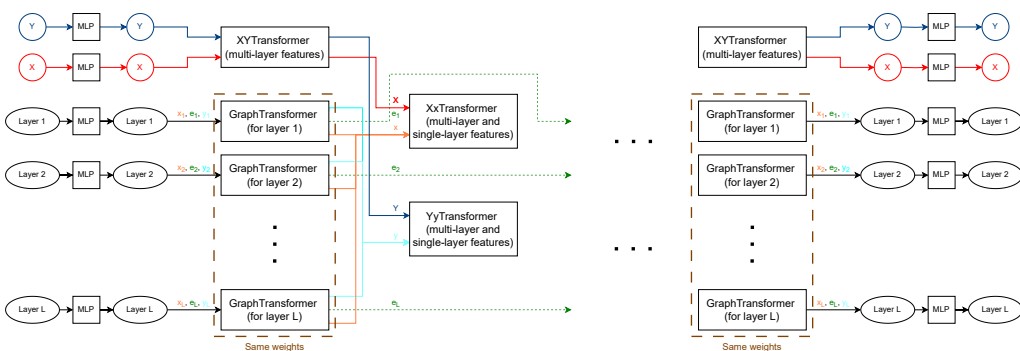

Figure 4: The entire architecture of the graph-superposition transformer

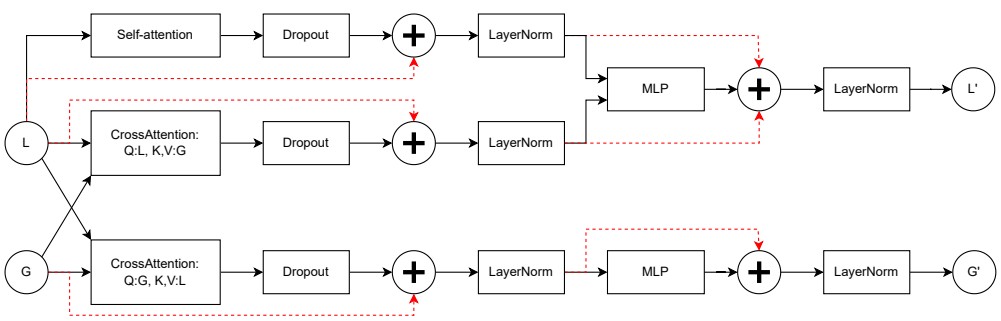

Figure 5: The cross-attention module between local features $\text{Ⓛ}$ and global features $\text{Ⓖ}$

## B    COMPLEXITY ANALYSIS

In the following, we estimate the asymptotic complexity of our proposed model, making use of the notation below:

- $n$: number of nodes
- $l$: number of layers in the graph-superposition representation
- $d$: dimension of the embeddings ($d = \max(d_x, d_e, d_y, d_X, d_Y)$)

We calculate the complexity relative to a batch containing a single hypergraph represented with a superposition of $l$ graphs with $n$ nodes and with embeddings of size $d$. This complexity applies to both a training step and a generation run. We will be considering every module of the graph-superposition transformer separately.

First, the graph-transformer block enables message passing between layer-specific node embeddings ($x$), edge embeddings ($e$), and graph embeddings ($y$). It contains linear modules on $x$, $e$, and $y$, with an asymptotic complexity of $O(n^2 l d)$ because there are $l$ layers with $n^2$ edge embeddings of size $d$ on each. These linear modules are here to calculate queries, keys, and values, but also to get the coefficients of the modulation factors in the Feature-Wise Linear Modulation (FiLM). The query-key product $\frac{QK}{\sqrt{d_f}}$ also has a cost of $O(n^2 l d)$, as well as the weighted sum of the attention head, the statistical pooling, or the FiLM on $e$ and $x$.

For the triplet aggregation, we only detail the inward case; the outward update has the same complexity. It relies on a two-step process:

- Calculation of the coefficients $a_{i,k}^{\text{in}} = \text{softmax}_k(b_{i,k}^{\text{in}})\sigma(g_{i,k}^{\text{in}})$
- Aggregation of these coefficients $o_{i,j}^{\text{in}} = \sum_{k=1}^{n} a_{i,k}^{\text{in}} v_{j,k}^{\text{in}}$

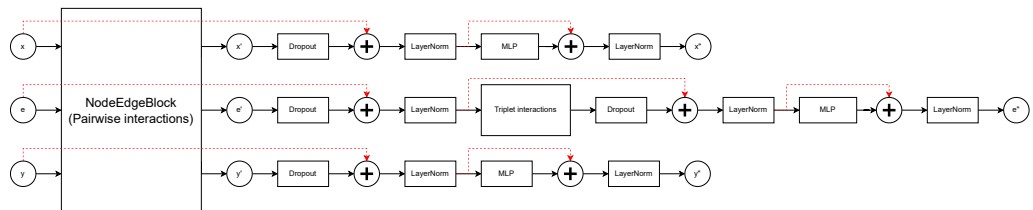

Figure 6: The integration of the triplet aggregation inside the graph transformer module

The first step has an asymptotic complexity of $O(n^2ld)$. The second step can be expressed as a matrix product:

$$O = AV^T$$

According to Hussain et al. (2024), this matrix product has a complexity of $\approx O(dn^{2.37})$ for a single layer, which gives a complexity of $\approx O(ldn^{2.37})$ for the entire graph superposition.

For the attention layer between $X$ and $Y$, the attention is calculated between $n$ multi-layer node embeddings and a global embedding, all of them of size $d$, which gives an asymptotic complexity of $O(n^2d)$. For the attention layer between $X$ and $x$, the attention is calculated, for each of the $n$ multi-layer node embeddings, between the multi-layer node embedding and the $l$ corresponding single-layer node embeddings, which gives a complexity of $O(l^2nd)$. For the attention layer between $Y$ and $y$, the attention is calculated between a global embedding and $l$ single-layer graph embeddings, which gives a complexity of $O(l^2n)$.

Before and after the transformer modules, there is, for every type of embedding, a multi-layer perceptron to do the conversion between the input/output representation and embeddings. If the ratio between the hidden dimensions and the dimensions of the embeddings is bounded, the complexity of these layers is $O(n^2ld)$.

Before passing the graph superposition to the network, the auxiliary information has to be calculated. Appendix A.1 provides two possible algorithms. We discuss the complexity for a single layer of the graph-superposition representation with $n$ nodes, $m_2$ edges, and $m_3$ 3-cliques in each algorithm.

The first basic algorithm consists of calculating the sums in Equation (2) with a product of elements in matrices. This operation is easy to parallelize and, for this reason, it is GPU-friendly. However, the complexity is $O(n^3)$ for the 3-cliques and $O(n^4)$ for the 4-cliques.

The second algorithm stops the nested loops when a couple of nodes in the set don't correspond to an edge. For the indices $i$ and $j$, all the possibilities are explored, but the index $k$ is examined if there is an edge between $i$ and $j$, which happens $m_2$ times. Then, if a 3-clique is detected, every possible fourth node is examined to check if it forms a clique with the three previous ones. For every 3-clique, these operations are in $O(n)$. Therefore, the complexity of the second algorithm for the cliques is in $O(n(n+m_2+m_3))$ for a single layer, and in $O(nl(n+m_2+m_3))$ for the entire graph superposition.

Consequently, the global complexity of the graph-superposition transformer is $\approx O(ldn^{2.37})$ without the auxiliary features, $\approx O(ldn^{2.37} + n^4l)$ with the first algorithm for the auxiliary information, and $\approx O(ldn^{2.37} + nl(n + m_2 + m_3))$ with the second one. However, even though the asymptotic factor in the complexity of the auxiliary information calculation is high, this process is often much faster than a forward pass in a neural network.

## C INTERMEDIATE EXPERIMENTAL RESULTS

We conducted experiments to compare our model with different variants of its architecture. The results for the variants are calculated from a set of 200 generated hypergraphs that are compared to the training dataset, where the results on our model are taken from the main experiments with 1000 generated hypergraphs.

Table 3 shows a comparison of the accuracy with different types of auxiliary features on the SBM dataset. Three variants are tested: a model without auxiliary features, a model with the auxiliary

| Experiment | Valid SBM | Node degree | Edge size | Spectral |
|---|---|---|---|---|
| No auxiliary feature | 0% | 0.558 | 0.0028 | 0.0072 |
| Cycle + spectral auxiliary features | 84% | 0.270 | 0.00352 | 0.00303 |
| Clique auxiliary features | 87.4% | 0.276 | 0.00273 | 0.00239 |

Table 3: Comparison of the accuracy of the graph-superposition transformer on the SBM dataset with different kinds of auxiliary features

| Experiment | Valid SBM | Node degree | Edge size | Spectral |
|---|---|---|---|---|
| Graph transformer on multi-hot encoded edge-labeled graphs | 68.5 % | 0.78 | 0.034 | 0.0091 |
| Graph-superposition transformer without triplet interaction | 77% | 0.563 | 0.0108 | 0.00419 |
| Graph-superposition transformer with triplet aggregation | 87.4% | 0.276 | 0.00273 | 0.00239 |
| Graph-superposition transformer with triplet attention | 86% | 0.416 | 0.00297 | 0.00294 |

Table 4: Comparison of different kinds of transformer architectures with clique auxiliary features on the SBM dataset

features used in Vignac et al. (2023) (cycle count and spectral auxiliary features), and our auxiliary features with cliques of size 3 and 4. The configuration without auxiliary data seems to be significantly less accurate than the others in all aspects of the generated hypergraphs. For the type of auxiliary features, there is no clear difference between those of Vignac et al. (2023) and ours regarding the proportion of valid SBM, the node degrees, and the Laplacian eigenvalues. However, the clique-specific auxiliary features achieve a better accuracy on the size of the hyperedges.

Table 4 shows a comparison of the accuracy of different variants of our architecture on the SBM dataset. The classic graph-transformer, when it is used with edge labels, gives a lower accuracy on every metric, which confirms the necessity to have a transformer model specifically designed for the topology of a graph superposition. Then, adding any form of triplet aggregation increases the accuracy of the model on every metric for this dataset. However, triplet attention, which induces heavier calculations, doesn't seem to improve the accuracy of the model compared to triplet aggregation, a lighter variant. For this reason, we decided to use triplet aggregation.

## D  EXPERIMENTAL DETAILS

### D.1  DATASETS

We tested our model on both synthetic and non-synthetic hypergraphs. These are the datasets used in Gailhard et al. (2025). Except for the hypertree dataset that contains 1000 hypergraphs for training, 100 for validation, and 100 for testing, these datasets contain 128 hypergraphs for training, 32 for validation, and 40 for testing.

**Erdos-Renyi hypergraphs**: This is the most basic way to generate random hypergraphs. It starts with a predefined set $S \subset \mathbb{N}^*$ of possible hyperedge sizes, and a family of probabilities $(p_i)_{i \in S}$. For a combination of $i \in S$ nodes in the hypergraphs $H = (\mathcal{V}, \mathcal{E})$, their probability to form a hyperedge is given by $p_i$:

$$\forall i \in S, \forall e \in \mathcal{P}_i(S), \mathbb{P}(e \in \mathcal{E}) = p_i$$

The dataset that is used is generated with $S = \{2; 3; 4\}$ and $p_2 = 0.1, p_2 = 0.005, p_4 = 0.0005$.

**Hypertrees**: First, a random tree with 32 nodes is sampled following a uniform distribution. Then, connected edges are randomly merged to form hyperedges with up to 5 nodes.

**SBM hypergraphs**: First, each of the 32 nodes is assigned to one of the two categories with independent and identically distributed Bernoulli processes of probability 0.5. Then, for every combination of 3 different nodes, the corresponding hyperedge is added to the hypergraph with probability

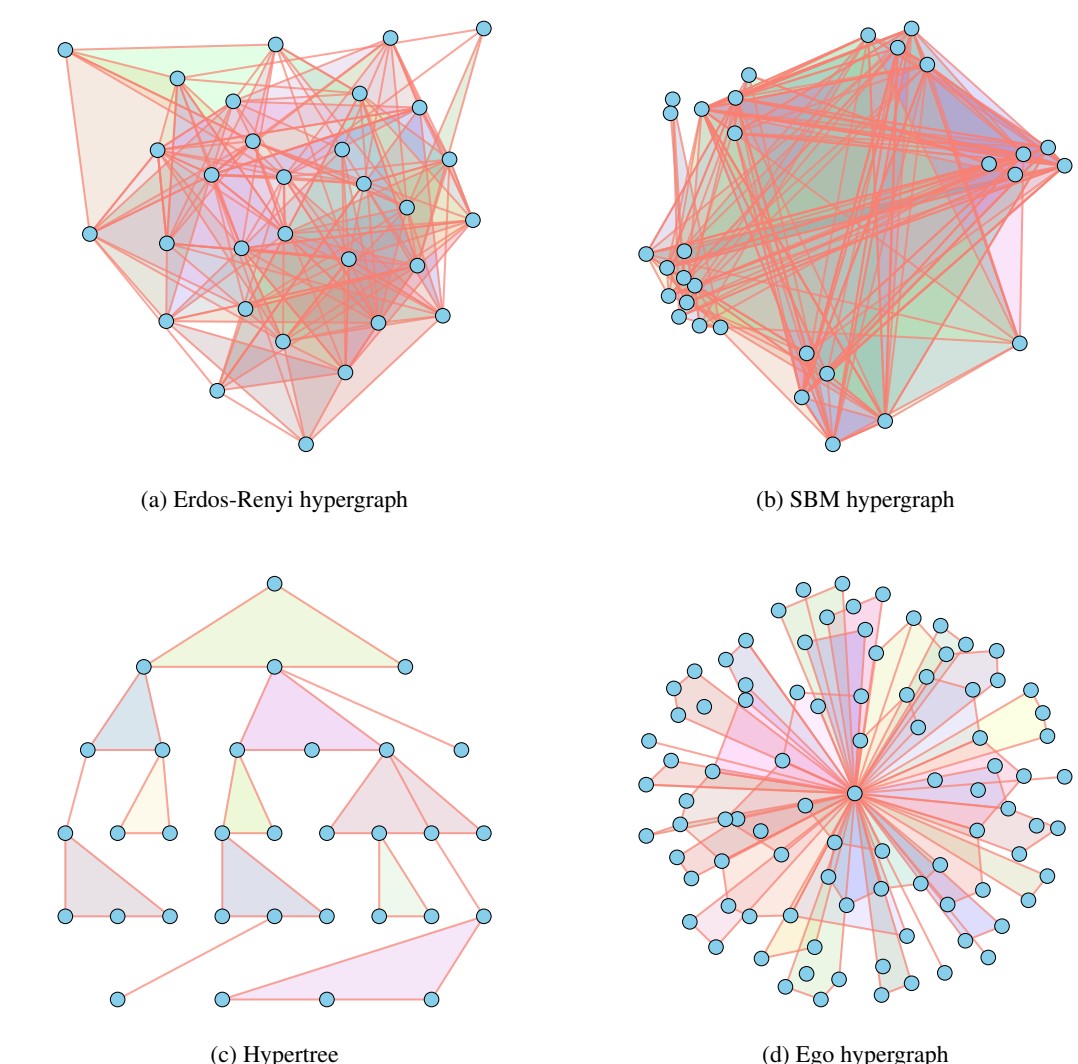

(a) Erdos-Renyi hypergraph

(b) SBM hypergraph

(c) Hypertree

(d) Ego hypergraph

Figure 7: Examples of hypergraphs from the synthetic datasets

0.05 if they are all in the same community, and with probability 0.001 if they belong to different communities.

**Ego hypergraphs**: First, a random number $n$ of nodes is chosen following a uniform distribution on $[\![150, 200]\!]$. Then 3000 random hyperedges are added with the following process: for each hyperedge, a random number $k$ of nodes is chosen uniformly in $[\![2, 5]\!]$. Then a random combination of $k$ nodes is chosen uniformly in $\mathcal{P}_k(\mathcal{V})$. Once all of the hyperedges have been added, a node is chosen uniformly in $\mathcal{V}$ to be the ego one. Finally, all of the hyperedges that don't contain the ego node are removed.

In all of the synthetic hypergraphs, even though there is a theoretical number of nodes, if a node is isolated, it is removed from the generated hypergraph. The Figure 7 provides an example for each type of synthetic hypergraph used in our experiments.

**ModelNet40 hypergraphs**: These datasets are made out of the conversion of the ModelNet40 datasets Wu et al. (2015) into hypergraph datasets for the classes *bookshelf*, *piano* and *plant*. These datasets contain only hyperedges of size 3.

## D.2 DESCRIPTION OF THE METRICS

The precision of our model and the baselines were evaluated using the metrics from Gailhard et al. (2025). Since HyperPA relies on a different generation process with a given distribution of node degrees and edge sizes for each hypergraph to be generated, we have kept the results of Gailhard et al. (2025) with only 40 generated hypergraphs. We also kept the results of Gailhard et al. (2025) for the Diffusion, VAE and GAN baselines because their poor precision made it difficult to calculate the metrics.

**Node degree**: The Wasserstein distance between the node degree distribution of the reference dataset and that of the generated batch. These distributions are made out of separately adding the degree of each node to a distribution that is common to all of the hypergraphs. This is not a comparison of the node degree repartition in the separate hypergraphs, but a comparison of the node degree distribution for the entire datasets.

**Edge size**: The Wasserstein distance between the edge size distribution of the reference dataset and that of the generated batch. As for node degrees, this is the distribution of the edge sizes across the entire dataset.

**Spectral**: The square of the Maximal Mean Discrepancy of the Laplacian eigenvalue distribution between the reference dataset and the generated batch.

**Node number**: For our experiments, it is the Wasserstein distance between the node number distributions of the reference dataset and the generated batch. For the values taken from Gailhard et al. (2025), it is the average difference between the target number of nodes and the number of nodes in the generated hypergraph.

**Validity metric for hypertrees**: The hypertrees are created by merging adjacent edges in a tree graph. They can be characterized by the fact that the only cycles in the clique expansion are subsets a hyperedges. This is the condition that is tested to compute the proportion of valid hypertrees in the generated batch.

**Validity metric for SBM hypergraphs**: The SBM hypergraphs are composed of two node communities with an intra-community and an inter-community probability to form a hyperedge. To determine whether a hypergraph is a valid SBM, the hypergraph is first projected into a graph using clique expansion. Then, an optimal community distribution is found. The hypergraph is considered a valid SBM if there are two communities and if the intra-community and extra-community probabilities correspond to the ones used to generate the training dataset.

**Validity metric for ego hypergraphs**: Since an ego hypergraph is a hypergraph containing one node belonging to all of the hyperedges, testing this property enables us to compute the proportion of valid ego hypergraphs in the generated batch.

**Closeness centrality, betweenness centrality and harmonic centrality**: These metrics come from Aksoy et al. (2020). They are calculated with $s = 1$.

**Uniqueness**: Proportion of generated hypergraphs that are not isomorphic to another generated one

**Novelty**: Proportion of generated hypergraphs that are not isomorphic to any hypergraph in the training dataset

## D.3 MULTI-LAYER GRAPH PROJECTION OF HYPERGRAPHS IN PRACTICE

As mentioned in Section 3.2, our algorithm to create a graph superposition with a hypergraph is not guaranteed to succeed. Indeed, if a conflict is detected between the maximal cliques of a layer so that they no longer represent the hyperedges of the initial hypergraph, the algorithm fails and is tried again with a different random seed until success. Using this process, we obtain a relatively low projection time compared to the model training time.

In Algorithm 2, each *MaximalCliques* call uses a variant of the Bron and Kerbosch algorithm Tomita et al. (2004), and has a time complexity of $O(3^{N/3})$. For $L$ layers and $N$ nodes, the total time complexity for the superposition projection is $O(3^{N/3}LN)$. However, most graphs in practice have a significantly lower number of maximum cliques than the theoretical maximum and are sparse, drastically reducing runtime Eppstein & Strash (2011). The max clique algorithm is shown to scale

to tens of thousands of nodes Tomita et al. (2004), leaving the denoising model, not projection, as the bottleneck of our system. The histograms in Figure 8 and Figure 9 show the runtime and the number of attempts until a successful projection for different datasets and layer counts.

Although our stochastic algorithm does not guarantee projection within the requested number of layers on the first attempt, we see that the majority of graphs are projected in under a second and require at most a handful of attempts. Only a small fraction of graphs require large numbers of attempts. Finally, when running Erdos-Renyi with a greater number of target layers, the number of attempts needed decreases as expected.

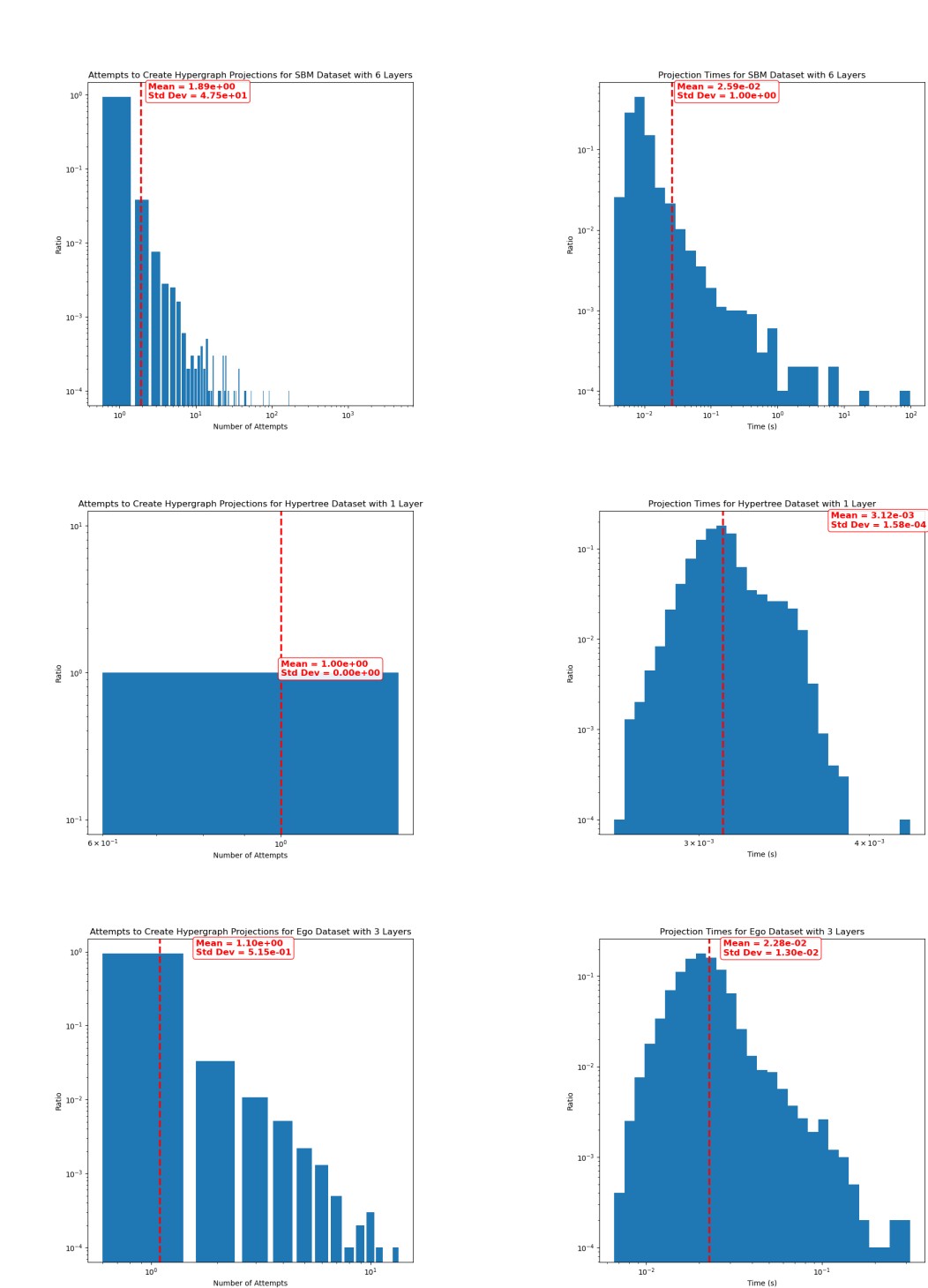

Figure 8: Histograms of number of attempts and calculation time for the projection of each of the 10 000 hypergraphs sampled from the distributions of SBM, Ego and Hypertree datasets

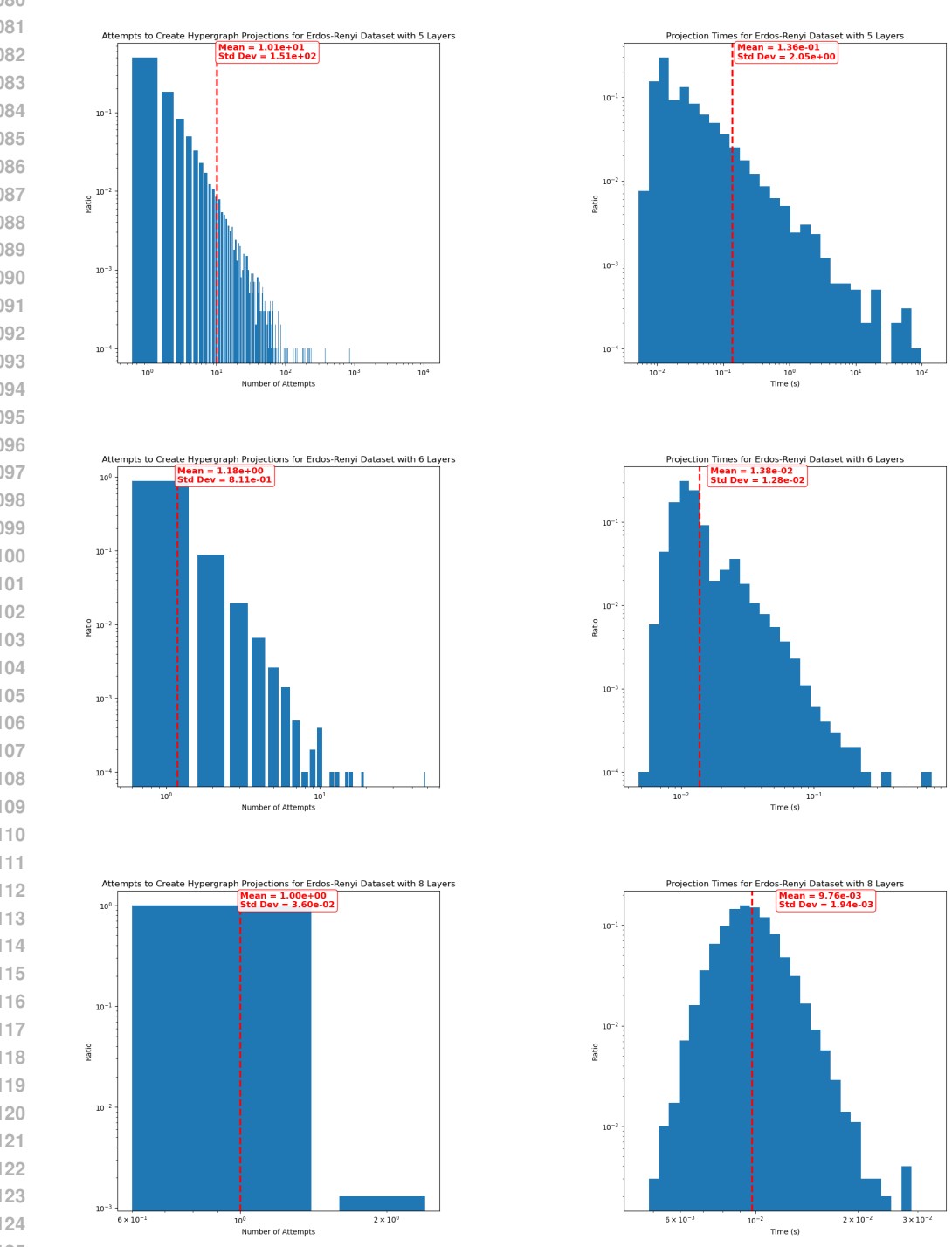

Figure 9: Histograms of number of attempts and calculation time for the projection of each of the 10 000 hypergraphs sampled from the distributions of the Erdos dataset with 5, 6 and 8 layers

## D.4 ADDITIONAL RESULTS

In Tables 5 to 7, we present the complete results of the experiments.

| | Erdos-Renyi Hypergraphs ($n_{avg}$ = 32, $std$ = 0.07) | | | | | | | | | SBM Hypergraphs ($n_{avg}$ = 31.73, $std$ = 0.55) | | | | | | | | | |
|---|---|---|---|---|---|---|---|---|---|---|---|---|---|---|---|---|---|---|---|
| Model | Node Num↑ | Node Deg↓ | Edge Size↓ | Spec-tral↓ | Uniq.↑ | Nov.↑ | Cent. Close↓ | Cent. Betw.↓ | Cent. Harm.↓ | Valid SBM↑ | Node Num↓ | Node Deg↓ | Edge Size↓ | Spec-tral↓ | Uniq.↑ | Nov.↑ | Cent. Close↓ | Cent. Betw.↓ | Cent. Harm.↓ |
| HyperPA | **0.0** | 5.5 | 1.8e-1 | 1.8e-1 | 1 | 1 | 7.8e-2 | 1.4e-2 | 1.1e2 | 2.5% | 7.5e-2 | 4.1 | 4.1e-1 | 2.7e-1 | 1 | 1 | 7.4e-2 | 8.0e-3 | 7.8e1 |
| VAE | 1.0e-1 | 2.1 | 5.4e-1 | 3.5e-2 | 1 | 1 | 7.9e-2 | 8.0e-3 | 1.4e1 | 0% | 3.8e-1 | 1.3 | 1.1 | 2.4e-2 | 1 | 1 | 7.0e-3 | 6.0e-3 | 6.5 |
| GAN | 6.8e-1 | 2.6 | 6.6e-1 | 4.8e-2 | 1 | 1 | 1.0e-1 | 1.1e-2 | 1.7e1 | 0% | 1.2 | 2.1 | 1.2 | 5.9e-2 | 1 | 1 | 7.6e-2 | 1.2e-2 | 1.1e1 |
| Diffusion | 5.0e-2 | 2.2 | 7.8e-1 | 1.4e-2 | 1 | 1 | 4.8e-2 | 3.0e-3 | 1.2e1 | 0% | 1.5e-1 | 1.7 | 1.4 | 3.1e-2 | 1 | 1 | 4.0e-2 | 4.0e-3 | 1.4e1 |
| HyperPLR | 1.7 (4e-2) | 5.6 (2e-1) | 1.3 (1e-2) | 1.2e-1 (2e-3) | 1 | 1 (0) | 1.4e-1 (3e-3) | **2.0e-7 (8e-9)** | 28 (9e-1) | 0 (0)% | 1.0e1 (1) | 1.4 (3e-2) | **0 (0)** | 4.0e-2 (2.1e-2) | 9.8e-1 (1e-2) | 1 (0) | 7.2e-2 (1.0e-2) | **2.1e-7 (2e-8)** | 1e1 (1) |
| HYGENE | 2.6e-2 | 2.0e-1 | 1.4e-1 | 4.7e-3 | 1 | 1 | 2.2e-2 | 5.6e-4 | 4.2 | 63% | **7.2e-2** | 1.7 | 3.5e-3 | 2.6e-2 | 1 | 1 | 1.6e-2 | 7.8e-3 | 1.2e1 |
| Ours | 7.2e-3 (1.3e-3) | **1.5e-1 (3e-2)** | **6.0e-3 (3.6e-3)** | **9.2e-4 (1.3e-4)** | 1 (0) | 1 (0) | **1.3e-3 (8e-4)** | 2.1e-4 (6e-5) | **1.2 (2e-1)** | **88.3% (0.7%)** | 1.0e-1 (1e-2) | **6.8e-1 (1.6e-1)** | 2.6e-3 (4e-4) | **3.1e-3 (2e-4)** | 1 (0) | 1 (0) | **5.1e-3 (3.2e-3)** | 2.2e-3 (7e-4) | 4.9 (1.3) |

Table 5: Detailed evaluation metrics for Erdos-Renyi and SBM hypergraphs. The values in parentheses represent the standard deviations over three experiments.

| | Ego Hypergraphs ($n_{avg}$ = 109.71, $std$ = 10.23) | | | | | | | | | | Tree hypergraphs ($n_{avg}$ = 32, $std$ = 0) | | | | | | | | | |
|---|---|---|---|---|---|---|---|---|---|---|---|---|---|---|---|---|---|---|---|---|
| Model | Valid Ego↑ | Node Num↓ | Node Deg↓ | Edge Size↓ | Spec-tral↓ | Uniq.↑ | Nov.↑ | Cent. Close↓ | Cent. Betw.↓ | Cent. Harm.↓ | Valid Tree↑ | Node Num↓ | Node Deg↓ | Edge Size↓ | Spec-tral↓ | Uniq.↑ | Nov.↑ | Cent. Close↓ | Cent. Betw.↓ | Cent. Harm.↓ |
| HyperPA | 0% | 3.6e1 | 2.6 | 4.2e-1 | 2.4e-1 | 1 | 1 | 3.5e-1 | 2.0e-3 | 1.4e2 | 0% | 2.4 | 3.2e-1 | 2.8e-1 | 1.6e-1 | 1 | 1 | 4.8e-1 | 1.7e-1 | 5.9 |
| VAE | 0% | 4.8e1 | 8.0e-1 | 1.5 | 1.3e-1 | 1 | 1 | 5.6e-1 | 1.9e-2 | 3.9e1 | 0% | 9.7 | 7.2e-2 | 4.8e-1 | 1.2e-1 | 1 | 1 | 2.8e-1 | 1.4e-1 | 3.9 |
| GAN | 0% | 6.0e1 | 9.2e-1 | 1.7 | 2.3e-1 | 1 | 1 | 6.1e-1 | 1.5e-2 | 4.2e1 | 0% | 6.0 | 1.5e-1 | 4.7e-1 | 8.9e-2 | 1 | 1 | 2.0e-1 | 1.2e-1 | 2.2 |
| Diffusion | 0% | 4.5 | 4.0 | 3.0 | 1.9e-1 | 1 | 1 | 4.1e-1 | 9.0e-3 | 6.9 | 0% | 2.2 | 1.7 | 1.9 | 1.3e-1 | 1 | 1 | 3.5e-1 | 1.4e-1 | 8.6 |
| HyperPLR | 0% | 1.3e1 | 4.7e-1 | 1.8e-1 | 2.2e-2 | 1 | 1 | 1.3e2 | **2.0e-7** | 5.4e2 | **100% (0%)** | **0 (0)** | **0 (0)** | **0 (0)** | **0 (0)** | 1 (0) | 0 (0) | **0 (0)** | **0 (0)** | **0 (0)** |
| HYGENE | **95.6%** | 9.3e-1 | 1.5e-1 | 4.5e-1 | 5.6e-3 | 1 | 1 | **1.2e-2** | 1.7e-4 | 3.4 | 79.2% | 0 | 3.2e-2 | 1.0e-1 | 7.9e-3 | 1 | 1 | 2.0e-2 | 1.7e-2 | 5.1e-1 |
| Ours | 50.6% | **3.1e-1** | **7.9e-2** | **8.0e-3** | **1.3e-3** | 1 | 1 | 1.5e-2 | 3.4e-4 | **1.6** | 90 (3.3)% | 4.0e-3 (1.7e-3) | 3.7e-3 (7e-4) | 3.2e-3 (4e-4) | 3.1e-4 (6e-5) | 1 (0) | 1 (0) | 6.4e-3 (3.2e-3) | 5.7e-3 (3.1e-3) | 9.3e-3 (3.3e-2) |

Table 6: Detailed evaluation metrics for Ego and Tree hypergraphs. The values in parentheses represent the standard deviations over three experiments.

| | ModelNet40 Bookshelf ($n_{avg}$ = 119.38, $std$ = 68.20) | | | | | | | | |
|---|---|---|---|---|---|---|---|---|---|
| Model | Node Num↓ | Node Deg↓ | Edge Size↓ | Spec-tral↓ | Uniq.↑ | Nov.↑ | Cent. Close↓ | Cent. Betw.↓ | Cent. Harm.↓ |
| HyperPA | 8.0 | 7.6 | 4.4e-2 | 4.8e-2 | 1 | 1 | 2.1e-1 | 5.0e-3 | 8.8e2 |
| VAE | 4.7e1 | 6.2 | 1.5 | 1.9e-1 | 1 | 1 | 1.5e-1 | 3.0e-3 | 1.1e2 |
| GAN | **0.0** | 4.0e2 | 4.6e1 | 4.8e-1 | 1 | 1 | 7.1e-1 | 7.0e-3 | 6.7e2 |
| Diffusion | **0.0** | 2.0e1 | 2.3 | 7.9e-2 | 1 | 1 | 2.4e-1 | 6.0e-3 | 2.6e2 |
| HyperPLR | 1.2 | **3.0** | **0** | **1.9e-3** | 1 | 1 | **2.8e-3** | **3.2e-7** | **5.6e1** |
| HYGENE | 1.7 | 8.2 | 3.1e-2 | 6.1e-2 | 1 | 1 | 8.0e-2 | 4.3e-3 | 2.0e2 |
| Ours | 2.4 | 3.2 | 7.6e-2 | 2.9e-2 | 1 | 1 | 1.6e-1 | 5.9e-3 | 9.0e1 |

Table 7: Detailed evaluation metrics for ModelNet40 Bookshelf

In Table 8, we present the training/sampling time and the maximal admitted batch size usage for HyperPLR, HYGENE, and SuperHype over different datasets. We use the maximum batch size admitted by each combination of model and dataset on our hardware as a measure of memory efficiency. We see that SuperHype is generally faster and consumes less memory compared to Gailhard et al. (2025), the closest, but still inferior, baseline in terms of quality. Wen & Yu (2025) is the most efficient model overall, but its generation quality is much lower compared to both other models.

| Dataset | Model | Sampling time (h:m:s) | Training time (h:m:s) | Max batch size |
|---|---|---|---|---|
| SBM | HYGENE | 00:42:20 | 19:00:00 | 20 |
| | HyperPLR | 00:00:08 | 00:02:12 | 128 |
| | SuperHype | 03:20:39 | 09:33:20 | 32 |
| Erdos | HYGENE | 01:11:37 | 23:00:00 | 20 |
| | HyperPLR | 00:00:14 | 00:02:29 | 128 |
| | SuperHype | 02:45:28 | 01:30:00 | 32 |
| Tree | HYGENE | 00:40:00 | 16:45:00 | 20 |
| | HyperPLR | 00:00:04 | 00:01:27 | 128 |
| | SuperHype | 00:27:24 | 27:53:26 | 200 |
| Ego | HYGENE | 05:26:40 | 16:00:00 | 10 |
| | HyperPLR | 00:00:25 | 00:08:11 | 128 |
| | SuperHype | 06:02:10 | 19:18:20 | 8 |

Table 8: Sampling/Training time and maximal batch size with SBM/Erdos/Tree/Ego dataset and HyperPLR/HYGENE/SuperHype

| Metric | Measurement | Erdos-Renyi | SBM | Hypertrees | Ego |
|---|---|---|---|---|---|
| Node deg. Wasserstein dist. | Distance between train and test | 0.0579 | 0.167 | 0.0215 | 0.0667 |
| | Predicted value for HYGENE | 0.475 | 0.321 | 0.059 | 0.063 |
| | Percentage of error | 12.2% | 52.0% | 36.4% | 105.9% |
| | Minimal value for HYGENE | 0.4171 | 0.154 | 0.0375 | 0 |
| | Maximal value for HYGENE | 0.5329 | 0.488 | 0.0805 | 0.1297 |
| Edge size Wasserstein dist. | Distance between train and test | 0.00655 | 0.0 | 0.0211 | 0.0522 |
| | Predicted value for HYGENE | 0.012 | 0.002 | 0.108 | 0.220 |
| | Percentage of error | 54.6% | 0% | 19.6% | 23.7% |
| | Minimal value for HYGENE | 0.00545 | 0.002 | 0.0869 | 0.168 |
| | Maximal value for HYGENE | 0.01855 | 0.002 | 0.1291 | 0.272 |
| Laplacian eigenvals. MMD | Distance between train and test | 0.00433 | 0.00486 | 0.00547 | 0.00119 |
| | Predicted value for HYGENE | 0.006 | 0.010 | 0.012 | 0.004 |
| | Percentage of error | 72.2% | 48.6% | 45.6% | 29.8% |
| | Minimal value for HYGENE | 0.000135 | 0.000917 | 0.00127 | 0.000827 |
| | Maximal value for HYGENE | 0.0205 | 0.0288 | 0.0337 | 0.00955 |

Table 9: Comparison of the distances between a batch generated with Gailhard et al. (2025) and a reference dataset and distances between the training dataset and the reference one

## E  MOTIVATION FOR CALCULATING METRICS WITH RESPECT TO TRAIN DATA

In Gailhard et al. (2025), the authors compute distances between the generated distribution and the distribution of a testing dataset. Even though both the training and testing datasets are sampled from the same probability distribution, the repartition of values such as node degree and edge size is not exactly the same.

If the distances between the sampled batch and the reference batch are much greater than those between the training dataset and the testing dataset, calculating the metrics with respect to the training dataset or the testing dataset doesn't induce a significant change in the final result. If not, the choice of the reference dataset will have a strong impact on the final metrics. Unlike Gailhard et al. (2025), we chose the training dataset.

First, if the training and the testing distributions are too different, even though they are sampled from the same probability distribution, it also means that the random process in which the testing dataset is sampled from the distribution has a significant impact on the final metrics. Indeed, the model is trained to match a probability distribution learned from the training dataset, so the bigger the distance between the training and the testing dataset is, the higher the distance between the testing dataset and the predicted batch is expected to be. Choosing the training dataset as a reference is a way to circumvent this random bias in validation metrics.

For example, for synthetic hypergraphs, the hyperedges are associated with a certain probability of existing, depending on their size and position. The model is expected to learn this distribution from the training dataset. It has no way to know the real distribution, and for this reason, it tries to reproduce the one from the training dataset. In this situation, comparing the predicted distribution with the one from the training dataset is the only correct way to measure the capability of the model to perform in this task.

Most of the time, having a testing dataset to evaluate a model is a way to know if the model is overfitting or not. It makes sense if we want to get the binary cross-entropy on a denoising task, but for generation, the model doesn't use any more data from any dataset. In that case, choosing a separate testing dataset doesn't give any more information on the model in itself, but it is a way to compare the bias that comes from the choice of the dataset with the one that comes from the imprecision of the model. To make sure that there is no overfitting, we added metrics on uniqueness and novelty for the generated batch.

On Table 9, distances between the training and the testing dataset are compared to the ones between the testing dataset and the sampled batch. Using these distances, the triangle inequality gives us upper and lower bounds for the distances between the training dataset and the sampled batch. For most cases, it is shown that the distance between the training and the testing dataset is responsible for most of the errors on the validation metrics when they are calculated with respect to the testing

dataset. It shows the need for a metric that truly reflects the precision of the model itself without being affected by random perturbations.

