# OpenReview forum: "SuperHype: Hypergraph Generation via Graph-Superposition Decomposition"
_ICLR.cc/2026/Conference — Submitted to ICLR 2026_

### Official Review · Reviewer_Ppc6 · 2025-10-31

**Soundness:** 3
**Presentation:** 3
**Contribution:** 3
**Rating:** 6
**Confidence:** 2

**Summary:**

The paper proposes SuperHype, a diffusion-based framework for hypergraph generation. The core idea is to represent a hypergraph as a graph superposition — a set of layered graphs that collectively preserve hyperedge information while keeping the representation tractable. The authors design a graph-superposition transformer to perform diffusion across and within these layers, augmented with auxiliary clique features and triplet aggregation to capture higher-order dependencies. Conceptually, the method reframes hypergraph synthesis as denoising over these layered graph projections rather than directly modeling hyperedges.

**Strengths:**

- The paper explores a relatively underexplored research direction in hypergraph generation.
- Overall, the manuscript is clearly written, and the design choices are well motivated.
- The empirical results are strong and support the proposed approach.

**Weaknesses:**

- The description of the Transformer (l. 306–316) is vague, relying on imprecise terms such as “mix” or “model interaction.” Figure 3 also fails to provide a clear structural overview of the model.
- In Table 1, it would be more informative to report the proportions of valid, unique, and novel graphs rather than only the valid ones, which would also resolve the asterisk issue for HyperPLR.
- The evaluation is limited to synthetic datasets, which does not strongly support the practical contribution of the work.
- Minor notation issues: $\mathcal{P}_2(e)$ is undefined, and the notation $\mathcal{V}$ and $\mathcal{E}$ seems to be used for both the vertex/edge sets and their cardinalities, causing ambiguity.
- In the definition of the loss, should $\mathcal{E}_l^t$ actually be $\mathcal{E}_l^0$?

**Questions:**

- The restriction to 3- and 4-cliques in the experiments is reasonable from a computational standpoint. However, the claim that this choice “covers most of the hyperedges (over more than two nodes)” across datasets should be supported by quantitative evidence for each dataset.
- Please report inference times to allow for a fair assessment of the method’s computational efficiency.
- Regarding the discussion on evaluation relative to the training or test dataset: have you considered simply generating more data? In practice, this would not necessarily increase training time, as a larger dataset can lead to faster convergence and reduced variability between training and test sets.

---

> ### Author Response · Authors · 2025-11-18
> **Reply to Reviewer Ppc6 - Part I**
>
> We thank the reviewer for the positive feedback. We appreciate the reviewer's description of our design choices as well-motivated and our empirical evidence as strong.
>
> We hereby answer Reviewer PPc6's comments. These comments further enable us to discuss the hypergraph transformer architecture and the ratios of Valid, Unique, and Novel generated hypergraphs. Additionally, we discuss the presence of large hyperedges in the datasets and report the sampling time of SuperHype.
>
> Thank you for the helpful comments!
>
> **W1: the decription of the graph-transformer is vague and Figure 3 does not provide a clear overview**
>
> terms and improve precision. To provide a better overview of the model and serve as a reference for our planned text changes, we include a streamlined Figure 3 via [anonymous link](https://i.postimg.cc/yNYdqT97/Fig-3-New.png). It exemplifies a fully noisy two-layer superposition $\mathcal V, (\mathcal E^T_1, \mathcal E^T_2)$ being denoised by the model. First, the `Hypergraph Layer Auxiliary Features` block creates initial node-, graph-, and edge-level representations from the graph structure for each layer. These embeddings then pass through the four components of a `Superposition Transformer` block. $\theta_\text{single}$ updates all per-layer embeddings independently. $\theta_\text{multi}$ creates an extra set of node- and graph-level embeddings from all layers. $\theta_\text{XMix}$ further updates the per-layer node-level embeddings, taking as input the concatenated node-related outputs of $\theta_\text{single}$ and $\theta_\text{XMix}$. $\theta_\text{YMix}$ behaves analogously to $\theta_\text{XMix}$, for graph-level embeddings.
>
> **W2: Table 1 should report the proportion of valid, unique, and novel graphs instead of only validity**
>
> We will replace 'Validity' with 'V.U.N' (Valid, Unique, Novel) in Table 1, and additionally report the three metrics separately in the tables of Appendix D.3, which currently report only uniqueness and novelty. As observed from the current versions of Tables 5, 6, and 7, uniqueness and novelty are nearly 100% across the board, except for HyperPLR Trees, where novelty is 0%. Please see the following [anonymous link](https://i.postimg.cc/DzBdP8v4/Tab-1-New.png), which previews the updated table for the upcoming paper revision.
>
> **W3: The evaluation is limited to synthetic datasets**
>
> We concur that datasets from a broader set of domains could better showcase the work's potential in practical applications. Unfortunately, as we were unable to identify suitable public datasets closely related to practical applications, we focused on evaluating our method using standard benchmark datasets also present in HYGENE, as the most relevant previous work. We note that, while still synthetic, ModelNet40 Bookshelf is motivated by a real application. The dataset contains featureless hypergraphs derived from point clouds.
>
> **W4: The manuscript has a few notation issues**
>
> We will make sure to explicitly define $\mathcal{P}_2(e)$, the set of all possible pairs of two nodes belonging to the same hyperedge, in the next manuscript version, in addition to clearing up the inconsistencies with $\mathcal{V}$ and $\mathcal{E}$.
>
> **W5: Should $\mathcal{E}^t_l$ be $\mathcal{E}^0_l$ in the loss definition?**
>
> Indeed, thank you for bringing it to our attention.
>
> **Q1: The claim that 3- and 4-cliques cover most of the hyperedges across datasets should be supported quantitatively**
>
> Our decision to consider only cliques of size 3 and 4 is indeed partly motivated by computational complexity. For an $N$-node graph, storing $k$-clique information involves storing a tensor with $N^k$ elements, and their computation requires temporarily computing and storing a matrix dot product of $N^{k+1}$ elements. The cost is generally feasible for 3- and 4-cliques, but not for 5-cliques or higher. In stating that 3- and 4-cliques "cover most of the hyperedges", we did not intend to imply that the hypergraphs contain mostly hyperedges of size 3 and 4. We instead meant to highlight that, for a larger maximal clique, every subset of size 3 and 4 is also a clique. In this situation, the larger clique is representable as a set of 3- and 4-cliques, which is sufficient to prevent most errors. Breaking up a large clique by omitting an edge also breaks up subcliques of size 3 and 4. As much is empirically verified, for instance, in the tree dataset, which contains hyperedges of size 5. We will rephrase the claim to improve clarity.

---

> > ### Author Response · Authors · 2025-11-18
> > **Reply to Reviewer Ppc6 - Part II**
> >
> > **Q2: Experiments do not report inference time**
> >
> > As highlighted within the answer to W4 of Reviewer UUqn, below, we include the training/sampling time and max admitted batch size usage for HyperPLR, HYGENE, and SuperHype over different datasets:
> > |Dataset|Model|Sampling time (h\:m\:s)|Training time (h\:m\:s)|
> > |-|-|-|-|
> > |SBM|HYGENE|00:42:20|19:00:00|
> > ||HyperPLR|00:00:08|00:02:12|
> > ||SuperHype|03:20:39|09:33:20|
> > |Erdos|HYGENE|01:11:37|23:00:00|
> > ||HyperPLR|00:00:14|00:02:29|
> > ||SuperHype|02:45:28|01:30:00|
> > |Tree|HYGENE|00:40:00|16:45:00|
> > ||HyperPLR|00:00:04|00:01:27|
> > ||SuperHype|00:27:24|27:53:26|
> > |Ego|HYGENE|05:26:40|16:00:00|
> > ||HyperPLR|00:00:25|00:08:11|
> > ||SuperHype|06:02:10|19:18:20|
> >
> > We see that SuperHype is generally faster compared to HYGENE, the closest, but still inferior, baseline in terms of quality. HyperPLR is the most efficient model overall, but its generation quality is much lower compared to both other models.

---

> > > ### Comment · Reviewer_Ppc6 · 2025-11-26
> > >
> > > I thank the reviewers for their clarifications and the additional figures.
> > >
> > > The efficiency study shows mixed results, although SuperHype is never significantly less efficient than the baselines.
> > >
> > > I continue to consider this a valuable contribution, that deserves acceptance. Given my limited experience with the particular topic of hypergraphs, and the study’s focus on synthetic datasets, I will nonetheless maintain my original score.

---

> > > > ### Author Response · Authors · 2025-11-26
> > > > **Acknowledgement to Reviewer**
> > > >
> > > > We thank the reviewer once more for their insightful comments and encouraging feedback.

---

### Official Review · Reviewer_hmho · 2025-11-01

**Soundness:** 2
**Presentation:** 3
**Contribution:** 3
**Rating:** 4
**Confidence:** 4

**Summary:**

This paper proposes SuperHype, a new diffusion-based model for generating hypergraphs. The key novelty is to represent a hypergraph as a graph superposition, which is a set of layered graphs whose maximal cliques together reconstruct all hyperedges. The authors design a Graph-Superposition Transformer for denoising the diffusion, and enhance it with clique-based auxiliary features and triplet aggregation. Experiments on five datasets show that SuperHype outperforms existing methods like HYGENE and HyperPLR on most metrics.

**Strengths:**

1. Generating realistic hypergraphs is an important topic to research.
2. The idea of breaking down distribute the hyperedges into the maximal cliques of multiple layered graphs, i.e. "graph superposition" is interesting and novel, although much further justification is needed -- see weaknesses below
3. Experiment design is overall reasonable; the results also show consistent improvement over baselines.

**Weaknesses:**

1. My key concern is lack of theoretical guarantee of the graph superposition projection process. For example, it would be very helpful to have theoretical insights about under what conditions the superposition projection algorithm can succeed (or fail); what's the relationship between the layer number d and the probability of success of the algorithm (and how do we choose a small enough d); are there many hypergraphs that fail the algorithm (which is closely tied to the general applicability of your method)

2. The denoising process also seems to lack enforcement/usage of some hard combinatorial constraints. For example, how can you guarantee that the generation result is (of high confidence) containing a set of "proper" maximal cliques. A counter example would be that you generate, say, a 10-complete graph with only 1 random edge missing. We know that in reality this structure is very unlikely the clique expansion of a real hypergraph. Also,  my sense is that different layers also have some hard combinatorial constraints with each -- I think it would be helpful to explicitly figure out what these constraints are, and how we can utilize them to help with denoising.

3. Typos: “The model ouperforms Gailhard et al. (2025)”: "ouperforms" should be “outperforms”; "an hypergraph" should be "a hypergraph"

**Questions:**

See weaknesses.

---

> ### Author Response · Authors · 2025-11-18
> **Reply to Reviewer hmho**
>
> We thank the reviewer for the thoughtful feedback. We appreciate that the reviewer recognizes SuperHype as an interesting and novel methodology.
>
> We hereby address Reviewer hmho's comments. We provide empirical evidence in support of the effectiveness of superposition projection and discuss design choices related to scaling down the combinatorial nature of hypergraphs.
>
> **W1: The graph superposition projection process has no theoretical guarantees**
>
> We acknowledge that our proposed hypergraph projection technique could benefit from a formulation that allows deriving greater theoretical insights. As an alternative to these insights, we would like to present an empirical analysis of the success rate of our superposition projection. As shared in the answer to W2 of Reviewer UUqn, we include, via anonymous links, **histograms** showing the **runtime** and **number of attempts** until a successful projection for different datasets and layer counts:
> |Dataset|Layers|Histograms|
> |-|-|-|
> |SBM|6|[Runtime](https://i.postimg.cc/6QYLfyW2/Hist-SBM-Time-6.png) / [#Attempts](https://i.postimg.cc/xdty3cfC/Hist-SBM-Attempts-6.png)|
> |Ego|3|[Runtime](https://i.postimg.cc/bv3H9sy8/Hist-Ego-Time.png) / [#Attempts](https://i.postimg.cc/q7kxK8Tq/Hist-Ego-Attempts.png)|
> |Tree|1|[Runtime](https://i.postimg.cc/pdqYJpWn/Hist-Tree-Time-1.png) / [#Attempts](https://i.postimg.cc/LsNB3n9q/Hist-Tree-Attempts-1.png)|
> |Erdos-Renyi|5|[Runtime](https://i.postimg.cc/5tnqS64t/Hist-ER-Time-5.png) / [#Attempts](https://i.postimg.cc/FKDgV7rN/Hist-ER-Attempts-5.png)|
> ||6|[Runtime](https://i.postimg.cc/JhPc5Grt/Hist-ER-Time-6.png) / [#Attempts](https://i.postimg.cc/zf0SFVzJ/Hist-ER-Attempts-6.png)|
> ||8|[Runtime](https://i.postimg.cc/bv3H9syw/Hist-ER-Time-8.png) / [#Attempts](https://i.postimg.cc/wB0cQ7q9/Hist-ER-Attempts-8.png)|
>
> Although our stochastic algorithm does not guarantee projection within the requested number of layers on the first attempt, we see that the majority of graphs are projected in under a second and require at most a handful of attempts. Only a small fraction of graphs require large numbers of attempts. Finally, when running Erdos-Renyi with a greater number of target layers, the number of attempts needed decreases as expected. We will add the analysis above to the appendix of the paper.
>
> **W2: The denoising process also lacks enforcement of hard combinatorial constraints**
>
> A potential formulation using hard combinatorial constraints could indeed further boost generation quality, especially for large hyperedges. From experimentally observed hyperedge size and node degree metrics, we find that our hypergraph-specific auxiliary features are nevertheless effective in guiding the model to create cliques. Furthermore, we can theoretically demonstrate that a small number of erroneous edges in the graph-superposition decomposition yields a limited difference in the final number of hyperedges. Take a graph with a single $N$-node maximal clique. If we remove a single edge from this clique, we get two maximal cliques of size $N-1$. Now, let $G$ be an arbitrary graph with an edge $e$ contained in $C$ maximal cliques. Also, let $G'$ be the graph $G$ without edge $e$. Based on the previous reasoning, we can upper bound the difference between the number of maximal cliques $M(G)$ in $G$ and the corresponding $M(G')$ in $G'$ as:
> $$ |M(G)-M(G')| \le C$$
> Similar reasoning can apply to an edge addition. Thus, even if the denoising process makes a few isolated errors on the edges, the result still corresponds to a hypergraph with a reasonable number of hyperedges. We plan to include a version of the above reasoning as an appendix in the next version of the manuscript.
>
> **W3: Manuscript has some typos**
>
> Thank you for the information. We have fixed the typos ahead of the following manuscript revision.

---

> > ### Author Response · Authors · 2025-11-28
> >
> > Dear Reviewer hmho,
> >
> > We hope our reply has helped address your questions and concerns. With the rebuttal period nearing its end, we would greatly appreciate it if you could inform us of any further aspects you would like us to address or update your assessment based on our existing replies.
> >
> > Kind regards,
> > The Authors

---

### Official Review · Reviewer_UUqn · 2025-11-01

**Soundness:** 2
**Presentation:** 3
**Contribution:** 3
**Rating:** 4
**Confidence:** 2

**Summary:**

The paper proposes a diffusion framework SuperHype for hypergraph generation. It decomposes a hypergraph into a small stack of ordinary layers, where maximal cliques in each layer map injectively to hyperedges. The paper borrows a graph-superposition transformer for within-layer and cross-layer message passing, augmented by hypergraph-specific features. Experiments show improved performance compared to baselines.

**Strengths:**

1. Hyper graph generation is a very classical yet important field. This paper tries to tackle this task via both modeling and theoretical way, which I feel is valuable.

2. Graph superposition achieves an injective mapping from layered cliques to hyperedges. If this condition could hold, the it should be a neat way to preserve high-order structure during duffision.

3. The paper integrates the discrete diffusion process with a graph superposition transformer, which looks novel to me.

**Weaknesses:**

1. Why is the number of hyperedges is N^2, not 2^{C(N, 2)} - 1?

2. In graph superposition projection, the paper claims that it uses a greedy algorithm to generate a graph superposition from a hypergraph in O(Ed). However, is the complexity for MaximalCliques in algo 1 linear?

3. The paper mentions memory cost and complexity analysis but does not conduct any experiments to justify this.

4. All the results from experimental session lack statistical significance. I would suggest to add error bound on them.

5. In Table 1, the proposed method does not yield the best results but still highlighted on Tree Hypergraphs.

**Questions:**

Please refer to my comments above.

---

> ### Author Response · Authors · 2025-11-18
> **Reply to Reviewer UUqn - Part I**
>
> We thank the reviewer for the thoughtful feedback. We appreciate the reviewer recognizing the paper's novelty and the relevance of the tackled problem.
>
> We hereby provide answers to Reviewer UUqn's comments. The comments enabled us to showcase SuperHype's efficiency under both memory and time costs, including the Maximal Clique subroutine. Finally, we provide statistical significance for our experiments and discuss the theoretical complexity of representing hypergraphs, as well as the superprojection representation.
>
> **W1: Why is the number of hyperedges $N^2$, not $2^{C(N, 2)} - 1$?**
>
> For $N$ nodes, a standard graph admits on the order of $O(N^2)$ edges, while a hypergraph admits $O(2^N)$ hyperedges. Our superposition projection transforms a hypergraph into $L$ (number of layers) classic graphs, each over the same $N$ nodes, leading to an $O(LN^2)$ combined edge count. We do not model interlayer edges between nodes of the same hypernode. To ensure coherent embedding between such related nodes, we harness the $\theta_\text{XMix}$ blocks in our superposition transformer architecture.
>
> **W2: The paper claims the superposition projection has $O(Ed)$ complexity; is the `MaximalCliques` complexity in Algorithm 1 linear?**
>
> The theoretical complexity in the paper indeed erroneously only accounts for the outer loops in Algorithm 1. Thank you for the opportunity to rectify the issue with the manuscript. Each `MaximalCliques` call uses a variant of the Bron and Kerbosch algorithm [1], and has a time complexity of $O(3^{N/3})$. For $L$ layers and $N$ nodes, the total time complexity for the superposition projection is $O(3^{N/3}LN)$. However, most graphs in practice have a significantly lower number of maximum cliques than the theoretical maximum and are sparse, drastically reducing runtime [2]. The max clique algorithm is shown to scale to tens of thousands of nodes [1], leaving the denoising model, not projection, as the bottleneck of our system. We include, via anonymous links, **histograms** showing the **runtime** and **number of attempts** until a successful projection for different datasets and layer counts:
> |Dataset|Layers|Histograms|
> |-|-|-|
> |SBM|6|[Runtime](https://i.postimg.cc/6QYLfyW2/Hist-SBM-Time-6.png) / [#Attempts](https://i.postimg.cc/xdty3cfC/Hist-SBM-Attempts-6.png)|
> |Ego|3|[Runtime](https://i.postimg.cc/bv3H9sy8/Hist-Ego-Time.png) / [#Attempts](https://i.postimg.cc/q7kxK8Tq/Hist-Ego-Attempts.png)|
> |Tree|1|[Runtime](https://i.postimg.cc/pdqYJpWn/Hist-Tree-Time-1.png) / [#Attempts](https://i.postimg.cc/LsNB3n9q/Hist-Tree-Attempts-1.png)|
> |Erdos-Renyi|5|[Runtime](https://i.postimg.cc/5tnqS64t/Hist-ER-Time-5.png) / [#Attempts](https://i.postimg.cc/FKDgV7rN/Hist-ER-Attempts-5.png)|
> ||6|[Runtime](https://i.postimg.cc/JhPc5Grt/Hist-ER-Time-6.png) / [#Attempts](https://i.postimg.cc/zf0SFVzJ/Hist-ER-Attempts-6.png)|
> ||8|[Runtime](https://i.postimg.cc/bv3H9syw/Hist-ER-Time-8.png) / [#Attempts](https://i.postimg.cc/wB0cQ7q9/Hist-ER-Attempts-8.png)|
>
> Although our stochastic algorithm does not guarantee projection within the requested number of layers on the first attempt, we see that the majority of graphs are projected in under a second and require at most a handful of attempts. Only a small fraction of graphs require large numbers of attempts. Finally, when running Erdos-Renyi with a greater number of target layers, the number of attempts needed decreases as expected. We will add the analysis above to the appendix of the paper.
>
> **W3: The paper lacks experimental measurements to justify the memory cost and complexity analysis**
>
> Below, we include the training/sampling time and max admitted batch size usage for HyperPLR, HYGENE, and SuperHype over different datasets:
> |Dataset|Model|Sampling time (h\:m\:s)|Training time (h\:m\:s)| Max batch size |
> |-|-|-|-|-|
> |SBM|HYGENE|00:42:20|19:00:00|20|
> ||HyperPLR|00:00:08|00:02:12|128|
> ||SuperHype|03:20:39|09:33:20|32|
> |Erdos|HYGENE|01:11:37|23:00:00|20|
> ||HyperPLR|00:00:14|00:02:29|128|
> ||SuperHype|02:45:28|01:30:00|32|
> |Tree|HYGENE|00:40:00|16:45:00|20|
> ||HyperPLR|00:00:04|00:01:27|128|
> ||SuperHype|00:27:24|27:53:26|200|
> |Ego|HYGENE|05:26:40|16:00:00|10|
> ||HyperPLR|00:00:25|00:08:11|128|
> ||SuperHype|06:02:10|19:18:20|8|
>
> We use the maximum batch size admitted by each combination of model and dataset on our hardware as a measure of memory efficiency. We see that SuperHype is generally faster and consumes less memory compared to HYGENE, the closest, but still inferior, baseline in terms of quality. HyperPLR is the most efficient model overall, but its generation quality is much lower compared to both other models.

---

> ### Author Response · Authors · 2025-11-18
> **Reply to Reviewer UUqn - Part II**
>
> **W4: Experiments lack statistical significance**
>
> As highlighted in the answer to **W2 & Q1** of Reviewer Nwfc, we rerun training and sampling with different seeds for three different datasets. We report below **mean and standard deviation values** over **three runs** for **V.U.N.** (valid, unique, novel) and **Wasserstein distance** metrics:
> |Dataset|Model|V.U.N.|Node Num|Node Deg|Edge size|Spectral|
> |-|-|-|-|--|--|--|
> |SBM|HyperPLR|0% (0%)|10 (0.1e1)|1.4 (0.03)|**0 (0)**|4.0e-2 (2.1e-2)|
> ||SuperHype|**88.3 % (0.7%)**|**1.0e-1 (0.1e-1)**|**6.8e-1 (1.6e-1)**|2.6e-3 (0.4e-3)|**3.0e-3 (0.2e-3)**|
> |Erdos-Renyi|HyperPLR|N/A|1.7 (0.04)|5.6 (0.2)|1.3 (0.006)|1.2e-1 (0.02e-1)|
> ||SuperHype|N/A|**7.2e-3 (1.3e-3)**|**1.5e-1 (0.3e-1)**|**6.0e-3 (3.6e-3)**|**9.2e-4 (1.3e-4)**|
> |Tree|HyperPLR|0% (0%)|0 (0)\*| 0(0)| 0(0)\*|0 (0)\*|
> ||SuperHype|**90% (3.3%)**|4.0e-3 (1.7e-3)|3.7e-3 (0.7e-3)|3.2e-3 (0.4e-3)|3.1e-4 (0.6e-4)|
>
> SuperHype's performance remains stable across not only sampling, but also training runs, and superposition projections. \*As for the main paper, note that HyperPLR's zero distance comes from merely outputting memorized training data. We omit HYGENE and the other datasets from the comparison in our current reply due to time constraints, and plan to integrate them into the next manuscript update.
>
> **W5: In Table 1, the proposed method does not yield the best results but still is highlighted on Tree Hypergraphs.**
>
> We highlighted SuperHype because, as noted in the table caption, HyperPLR technically has higher validity, but it merely copies training graphs. However, we agree that this is unnecessarily confusing. Therefore, in accordance with Reviewer Ppc6's suggestion, we will replace 'Validity' with 'V.U.N' (Valid, Unique, Novel) in Table 1, and additionally report the three metrics separately in the tables of Appendix D.3.
> Please see the following [anonymous link](https://i.postimg.cc/DzBdP8v4/Tab-1-New.png), previewing with the updated table to be included in the following paper revision.
>
> [1] Tomita, E., Tanaka, A., & Takahashi, H. (2006). The worst-case time complexity for generating all maximal cliques and computational experiments. Theoretical computer science, 363(1), 28-42.
>
> [2] Eppstein, D., Löffler, M., & Strash, D. (2010, December). Listing all maximal cliques in sparse graphs in near-optimal time. In International symposium on algorithms and computation (pp. 403-414). Berlin, Heidelberg: Springer Berlin Heidelberg.

---

> > ### Author Response · Authors · 2025-11-28
> >
> > Dear Reviewer UUqn,
> >
> > We hope our reply has helped address your questions and concerns. With the rebuttal period nearing its end, we would greatly appreciate it if you could inform us of any further aspects you would like us to address or update your assessment based on our existing replies.
> >
> > Kind regards,
> > The Authors

---

### Official Review · Reviewer_Nwfc · 2025-11-01

**Soundness:** 3
**Presentation:** 3
**Contribution:** 3
**Rating:** 8
**Confidence:** 3

**Summary:**

The paper derives a novel decomposition of hyper graphs  into layered clique decompositions which preserves the graph exactly while retaining the compactness of the clique decomposition (at the cost of not necessarily always existing). The authors then present an  adapted architecture for it based on the Digress GAT, adding sharing mechanisms that allow information to flow between the layers of the decomposition, as well as adding auxillary features constructed specifically for the hypergraph case . The method is evaluated on ER,SBM,tree and ego-hypergraphs, against HYGENE and its baselines, as well as HyperPA and HyperPLR. the framework is presented as overall competetive/best in class across the datasets

**Strengths:**

1. originality: resolving the tractability/exactness problem of hypergraph to graph embedding via effectively random projections similar to how sliced wasserstein approximations work  (using the terms a bit loosely) is a clever idea, even if it comes without guarantees

2. quality: evaluation done mainly rigorously

3. clarity: well written and legible

4. significance: strong improvements on most datasets

**Weaknesses:**

- line 41 “These recently proposed hypergraph synthesizers, albeit they are based on architectures that are unfit for hypergraphs’ characteristics, and bring limited generative capabilities.” parses weirdly, seems to be some editing leftover
- would like to see multiple seeds, CIs (stochasticity in the decompositoin could blow up variance in performance)

**Questions:**

- address the CI issue please
- can the different layers be modeled as a single big graph? why and why not? (scaling I assume? )

---

> ### Author Response · Authors · 2025-11-18
> **Reply to Reviewer Nwfc**
>
> We thank the reviewer for the positive feedback. We appreciate the reviewer's acknowledgement of the significance and originality of our work.
>
> We hereby answer Reviewer Nwfc's questions. We provide confidence intervals that demonstrate the stability of SuperHype performance and clarify the need for the superprojection representation in hypergraphs.
>
> **W1: A sentence parses weirdly**
>
> We have replaced the mistyped sentence in a revised manuscript with
> > These recently proposed hypergraph synthesizers rely on architectures that are unsuitable for the characteristics of hypergraphs and offer limited generative capabilities.
>
> **W2 & Q1: Evaluation lacks confidence intervals across multiple seeds**
>
> We rerun training and sampling with different seeds for three different datasets. We report below **mean and standard deviation values** over **three runs** for **V.U.N.** (valid, unique, novel) and **Wasserstein distance** metrics:
> |Dataset|Model|V.U.N.|Node Num|Node Deg|Edge size|Spectral|
> |-|-|-|-|--|--|--|
> |SBM|HyperPLR|0% (0%)|10 (0.1e1)|1.4 (0.03)|**0 (0)**|4.0e-2 (2.1e-2)|
> ||SuperHype|**88.3 % (0.7%)**|**1.0e-1 (0.1e-1)**|**6.8e-1 (1.6e-1)**|2.6e-3 (0.4e-3)|**3.0e-3 (0.2e-3)**|
> |Erdos-Renyi|HyperPLR|N/A|1.7 (0.04)|5.6 (0.2)|1.3 (0.006)|1.2e-1 (0.02e-1)|
> ||SuperHype|N/A|**7.2e-3 (1.3e-3)**|**1.5e-1 (0.3e-1)**|**6.0e-3 (3.6e-3)**|**9.2e-4 (1.3e-4)**|
> |Tree|HyperPLR|0% (0%)|0 (0)\*| 0(0)| 0(0)\*|0 (0)\*|
> ||SuperHype|**90% (3.3%)**|4.0e-3 (1.7e-3)|3.7e-3 (0.7e-3)|3.2e-3 (0.4e-3)|3.1e-4 (0.6e-4)|
>
> SuperHype's performance remains stable across not only sampling, but also training runs, and superposition projections. \*As for the main paper, note that HyperPLR's zero distance comes from merely outputting memorized training data. We omit HYGENE and the other datasets from the comparison in our current reply due to time constraints, and plan to integrate them into the next manuscript update.
>
> **Q1: Can the different layers be modeled as a single big graph?**
>
> Having a single graph with all layers is indeed possible, but it has scaling-related issues. Representing a graph with N nodes using the adjacency matrix results in $O(N^2)$ memory complexity. Thus, for $L$ layers, having layers as subgraphs of a larger graph would have complexity $O(L^2N^2)$. Allowing information exchange between projection nodes belonging to the same hypergraph node would require, for instance, adding extra edges to the graph. Our representation involves L adjacency matrices of graphs with N nodes, giving a lower complexity of $O(LN^2)$, while $\theta_\text{XMix}$ and $\theta_\text{YMix}$ in our graph-superposition transformer account for different projection nodes belonging to the same hypergraph node.

---

> > ### Comment · Reviewer_Nwfc · 2025-11-25
> > **I thank the authors for their rebuttal**
> >
> > Thank you, i think the already strong papers is further strengthened by this in my eyes

---

> > > ### Author Response · Authors · 2025-11-26
> > > **Acknowledgement to Reviewer**
> > >
> > > We would like to thank the reviewer once again for their valuable feedback and for the positive comments regarding our rebuttal.

---

### Author Response · Authors · 2025-11-18
**Rebuttal Summary**

Dear Reviewers,

Thank you for your insightful comments!

Below is a summary of the key updates we have made to address your concerns.

New experiments:
- Reviewers _Nwfc_ and _UUqn_: Run sampling and training with multiple seeds and report confidence interval
- Reviewers _UUqn_ and _Ppc6_: Add comparison of SuperHype's runtime (and memory usage) compared to baselines
- Reviewers _UUqn_ and _hmho_: Provide analysis on runtime and attempts until success of superposition projection

We will include all new experiments and related analysis in the next version of the manuscript.

Additional updates:
- Reviewer _Nwfc_: clarify the advantage of modelling layers via separate graphs
- Reviewer _UUqn_: clarify the number of possible edges in hypergraphs and in projection layers; fix the erroneous in superposition projection complexity
- Reviewer _hmho_: provide additional context on the denoising error in our current denoising formulation
- Reviewer _Ppc6_: create a new streamlined Figure 3; replace the ratio of valid graphs with valid, unique, and novel ones in the main paper tables; clarify the claim of 3- and 4-cliques.

---

### Author Response · Authors · 2025-11-24
**Updated manuscript addressing your feedback**

Dear Reviewers,

We have carefully addressed all points raised in your reviews and have updated the manuscript accordingly. The main changes include:
- An update of the results tables to add averages and standard deviations for different seeds in the main experiments
- The addition of training and evaluation times, as well as memory consumption
- An analysis of the projection in terms of computation time and number of attempts

We have also corrected a few errors and clarified some ambiguous passages. All modifications are highlighted in the revised manuscript for easy reference.
We hope that these corrections improve the paper and address your concerns. We would greatly appreciate your thoughts on our responses and the updated version. If any points require further clarification or if you have additional questions, we are happy to provide more details.

---

### Meta-Review · Area_Chair_1WeL · 2026-01-07

**Summary:**

The paper introduces the first diffusion framework, i.e., SuperHype for hypergraph generation. The main component is the Graph-Superposition Transformer which treats the superposition as an interconnected sequence of layers. Experimental results on five datasets display competitive results of SuperHype. However, the main issues lie in the limitation of theoretical guarantee and explanations of results. Some reviewers raised concerns about memory usage/comparison and significance of results. The responses from authors did not fully address these concerns.

**Reviewer Concerns:**

Part of concerns and comments are addressed by authors.

**Reviewer Scores:**

They might keep same scores.

---

### Decision · Program_Chairs · 2026-01-26

Reject